



# Energy states of soil water – a thermodynamic perspective on storage dynamics and the underlying controls

Erwin Zehe[1], Ralf Loritz[1], Conrad Jackisch[1], Martijn Westhoff[2], Axel Kleidon[3],

Theresa Blume[4], Sibylle Hassler[1], Hubert, H. Savenije[5]

1) Karlsruhe Institute of Technology (KIT), 2) Vrije Universiteit Amsterdam, The Netherlands, 3),

Max Planck Institure for Bio-Geo-Chemistry, Jena 4), GFZ German Research Centre for

Geosciences 5) Delft Technical University.

Abstract: The present study corroborates that the free energy state of soil water offers a new perspective on storage dynamics and similarity of hydrological systems that cannot be inferred from the usual comparison of soil moisture observations or groundwater levels. We show that the unsaturated zone of any hydrological system is characterized by a system-specific balance of storage and release. This storage equilibrium, which is jointly controlled by the soil physical and topographical system characteristics, reflects the thermodynamic equilibrium state of minimum free energy the system approaches when relaxing from external disturbances. Rainfall or radiation frequently forces parts of the system out of this storage equilibrium, storage dynamics can hence be visualized as sequences of deviations from and relaxations back to equilibrium. This perspective reveals that storage dynamics operates in two distinctly different energetic regimes, where either capillarity dominates over gravity or vice versa. As these regimes are associated either with a storage deficit or a storage excess, relaxation requires either recharge or release. This implies that the terms 'wet' and 'dry' should be used with respect to the equilibrium storage as meaningful reference point. We show furthermore that the free energy state of the soil water stock, the storage equilibrium which separates the two dynamic regimes, as well as the degree of non-linearity within those regimes depend on the joint controls of catchment topography and the physical properties of the soils. We express these joint controls in form of a new characteristic function of the unsaturated zone we call the 'energy state function'. By comparing the energy state functions of different systems we demonstrate their distinct sensitivity to topography and soil water characteristics and their usefulness for inter-comparing storage dynamics among those systems. This ultimately reveals that storage dynamics at the system level may operate by far more linearly than suggested by the retention function of the soils.





# 1  INTRODUCTION

## 1.1  Motivation

'The whole is greater than the sum of the parts' - Savenije and Hrachowitz (2017) grounded their recent proposition that catchments function similarly to meta-organisms on this famous quote of Aristotle. Their blue print essentially suggests that catchments evolve towards a configuration which balances water storage and release in an optimal manner. This idea is largely motivated by their more specific finding of an optimum rooting depth (Gao et al., 2014), which likely balances the advantage of vegetation to endure droughts of increasing return periods with the necessary energetic investment to grow their roots to deeper and deeper water stocks. The present study revisits the idea that hydrological systems balance storage and release suggested by Savenije and Hrachowitz (2017), using a thermodynamic perspective on soil water dynamics (Zehe et al., 2013). More specifically we propose that this balance connects to the thermodynamic equilibrium state the system approaches when relaxing from external disturbance driven by either rainfall or radiative forcing.

## 1.2  Thermodynamic reasoning in hydrology

Thermodynamic reasoning in earth sciences may be traced back to the early work of Leopold and Langbein (1962) on the role of entropy in the evolution of landforms. Thermodynamics gained however substantial attention in catchment hydrology since the work of Kleidon and Schymanski (2008). Kleidon and Schymanski (2008) discussed the opportunity of using thermodynamic optimality such as maximum entropy production (MEP, Paltridge, 1979) or maximum power (Lotka, 1922a; Lotka, 1922b) for uncalibrated hydrological predictions. This vision has motivated several efforts to predict the catchment water balance using MEP either to determine parameters controlling root water uptake (Porada et al., 2011) or to optimize the splitting of rainfall into recharge and (surface) runoff (Westhoff and Zehe, 2013; Zehe et al., 2013). Other studies investigated the role of connected flow networks such as river networks or rill systems and suggested that they increase the power in coupled water and sediment fluxes (Howard, 1990; Favis-Mortlock et al., 2000; Paik and Kumar, 2010; Kleidon et al., 2013). This is because these networks minimize local dissipative losses for instance in the river network (Rinaldo et al., 1996) or in subsurface preferential flow paths (Zehe et al., 2010; Hergarten et al., 2014). Recent studies employed thermodynamic optimality approaches to predict partitioning of net short wave radiation into long wave outgoing radiation and turbulent fluxes of latent and sensible heat (Kleidon et al., 2014; Renner et al., 2016), to





derive the Budyko curve (Wang et al., 2015; Westhoff et al., 2014; Westhoff et al., 2016), to
explain root water uptake (Hildebrandt et al., 2016) or to infer parameters controlling salt
water intrusion into estuaries (Zhang and Savenije, 2018).

While the potential of thermodynamic optimality for uncalibrated predictions is an exciting
issue, a thermodynamic perspective alone has a lot to offer to hydrological sciences. For
instance it can be used to explain hydrological similarity based on a thermodynamically
meaningful combination of catchment characteristics (Zehe et al., 2014; Seibert et al., 2017;
Loritz et al., 2018). Or it motivated the effort to develop models of intermediate complexity,
for instance based on the idea of a representative elementary watershed REW (Reggiani et al.,
1998a; Reggiani et al., 1998b; Reggiani et al., 1999; Reggiani et al., 2000; Reggiani and
Schellekens, 2003; Lee et al., 2005; Zhang et al., 2005; Tian et al., 2006; Lee et al., 2007;
Sivapalan, 2018). Closely related to this, thermodynamic reasoning has also been used to
upscale effective soil water characteristics (Zehe et al., 2006; de Rooij, 2009) partly for
closure of the REW approach. In this study we propose that thermodynamic reasoning offers a
radically new, energy based perspective on storage dynamics and similarity of hydrological
systems that cannot be inferred from the usual comparison of soil moisture observations or
groundwater levels.
**1.3   The 'energy perspective' on soil water storage**
In line with Savenije and Hrachowitz (2017) we propose that the unsaturated zone of any
hydrological system is characterized by a system-specific balance of storage and release. This
balance, which is jointly controlled by the soil physical and topographical characteristics,
relates to the thermodynamic equilibrium of the system, as it corresponds to a state of
minimum free energy of the soil water stock. In the absence of an external rainfall or radiative
forcing, the system will thus naturally relax back to this storage equilibrium and remain in this
state. Hydrological systems are however not isolated, which implies that they are frequently
forced out of their equilibrium either by rainfall or by radiation (Fig. 1). Here we show that
storage dynamics can be visualized as deviations of the free energy state of soil water from
this storage equilibrium. This reveals that these deviations and subsequent relaxations operate
in two distinctly different energetic regimes, which are associated with either with a storage
excess or a storage deficit relative to the equilibrium state. Radiation driven evaporation and
transpiration force the system out of its equilibrium into a state range where capillarity
dominates against gravity, or in energetic terms, capillary surface energy of soil water is in



absolute terms larger than its potential energy. We thus call this the "C-regime", because
capillary surface energy differences act as dominant driver of soil water dynamics. The
system is in a state of a storage deficit as it needs to recharge water for relaxation, but the
necessary recharge amount is determined by the energetic distance to equilibrium. In contrary,
rainfall driven recharge pushes the system into a state range where gravity dominates against
capillarity, or in energetic terms, capillary surface energy of soil water is in absolute terms
smaller than its potential energy. We call this the 'P-regime' because soil water dynamics is
predominantly controlled by potential energy differences. The system is in a state of storage
excess. It needs to release water to relax back to equilibrium and the necessary amount is
determined by its energetic distance from equilibrium.

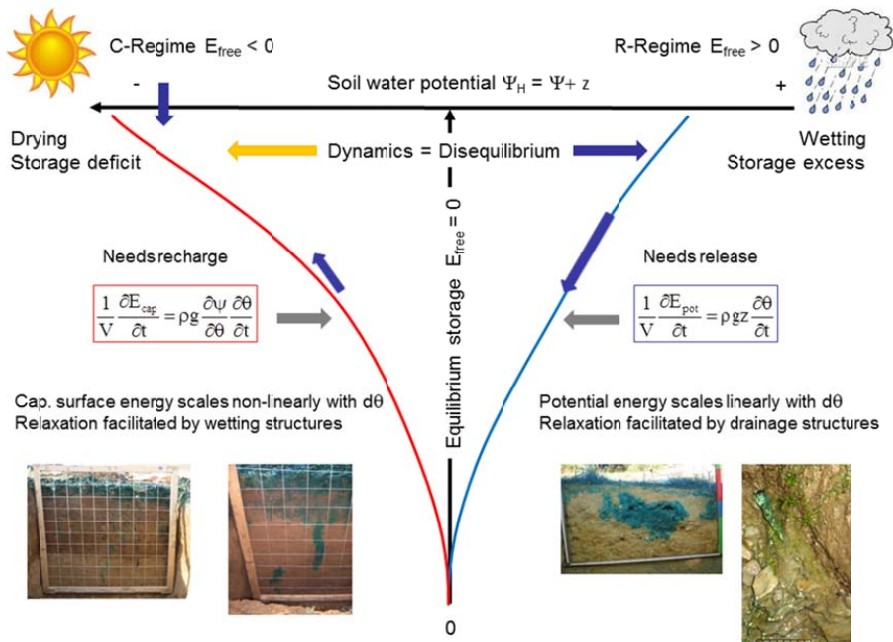


Figure 1 (from Zehe et al (2013), modified): Thermodynamic equilibrium in a soil profile above the
ground water surface as state of zero total hydraulic potential. Drying pulls the system into the C-
regime by increasing absolute values of negative capillary surface energy. Wetting pushes the system
into the P-regime by increasing potential energy of soil water. The expression for capillary surface
energy and potential energy of soil water and their dynamic change with the soil water content are
derived in section 2.1. The brilliant blue image highlight that relaxation back to equilibrium is in both
regimes facilitated by different types of preferential pathways, those shown in the left favor recharge
those shown to the right favor release.





As further detailed in the discussion section, relaxation back to equilibrium and thus
dissipation of free energy is in both regimes accelerated by preferential pathways, which
either favor recharge of the dry soil matrix to deplete the storage deficit or release of water to
deplete the storage excess (Zehe et al. 2013).
### 1.4 Objectives
In the following we show that the free energy state of the soil water stock, the distribution of
equilibrium storage values in a system, as well as the degree of non-linearity within the
aforementioned regimes depend on the joint controls of catchment topography, the
groundwater surface and the physical properties of the soils. These joint controls can be
expressed in form of a new characteristic function, which relates free energy of soil water to
a) the relative saturation of the soil, b) the corresponding matric/soil water potential and c) the
topographic elevation above groundwater. As this function characterizes the possible range of
"energy states" of soil water stored in the system, we call it the vadose zone "energy state
function". By comparing the energy state functions of different systems we demonstrate their
distinct sensitivity to topography and soil water characteristics. We show furthermore that soil
water dynamics at single plots or within an entire hydrological system can, in case of a slowly
varying groundwater table, be nicely visualized as deviations from the storage equilibrium
either into the P- or the C-regime and subsequent relaxation due recharge or release of water.
This offers new opportunities for inter-comparing storage dynamics among different systems,
to explain differences in the corresponding of runoff generation and to which degree the point
scale non-linearity of soil physical properties affects storage dynamics at the system level.
## 2 THEORY
In the following we express the drivers of soil water dynamics, the soil water or matric
potential and the gravity potential, in energetic terms and then derive the energy state
function. As the latter depends on the elevation above the groundwater surface and the
retention function of the soils, we present those energy state functions for observed soil water
retentions from different landscapes to illustrate its sensitivity to those factors.
### 2.1 Free energy of the soil water
Latest since the work of Iwata et al. (1995) it is known that energetic state of water stored in
unsaturated soil depends on its potential energy and the surface energy at the air-water
interface. We may hence express the change in Helmholtz free energy (J) of the amount of



water stored in a small control volume dV ($m^3$) based on the changes in its potential energy
and of the surface energy at the air-water-interface (Iwata et al., 1995) [1]:

$$dE_{free} = gz_{GW}dM + \sigma dA \quad \text{(Eq. 1)}$$

Where g ($ms^{-2}$) is the acceleration of the earth and dM (kg) denotes a change in the stored
water mass, $z_{GW}$ (m) is the depth above the groundwater surface, $\sigma$ (N/m) is the surface
tension of water and dA ($m^2$) the change in the area of the water air interface. When
expressing dM as product of the water density $\rho$ ($kgm^{-3}$) and a change in the volume of the
water phase $dV_\theta$ ($m^3$) we obtain:

$$\text{a) } dE_{free} = \rho g z_{GW} dV_\theta + \sigma dA \Leftrightarrow$$
$$\text{b) } \frac{\partial E_{free}}{\partial V_\theta} = \rho g z_{GW} + \sigma \frac{\partial A}{\partial V_\theta} \qquad \text{Eq. (2)}$$


Particularly Eq. 2b) highlights that a change in the volume of the water phase implies, on one
hand a change in its potential energy. On the other hand it leads to changes in the surface
energy, as the air-water-interface and its curvature change with changing soil water content as
well. In the next step we employ the definition of the soil water potential $\psi$ (m) for a spherical
air-water-interface with curvature radius r (m) to eliminate the surface tension in Eq. 2:

$$\psi = \frac{2\sigma}{\rho g r} \Leftrightarrow \sigma = \frac{r}{2}\rho g \psi(\theta) \quad \text{Eq. (3)}$$

This yields the following expressions to characterise the change in free energy as function of a
changing volume of the water phase:

$$\text{a) } dE_{free} = \rho g z_{GW} dV_\theta + \frac{r}{2}\rho g \psi(\theta)\frac{\partial A}{\partial V_\theta} dV_\theta \Leftrightarrow$$
$$\text{b) } \frac{\partial E_{free}}{\partial V_\theta} = \rho g z_{GW} + \frac{r}{2}\rho g \psi(\theta)\frac{\partial A}{\partial V_\theta} \qquad \text{Eq. (4)}$$


---

[1] Note that we assume isothermal conditions and neglect volumetric changes of the pore space.



174 Note the change rate in surface area of a sphere with changing radius and the related change

175 rate in volume are as follows:


$$a) \frac{\partial A_\theta}{\partial r} = 8\pi r \vee \frac{\partial V_\theta}{\partial r} = 4\pi r^2 \Leftrightarrow$$
$$b) \frac{\partial A_\theta}{\partial V_\theta} = \frac{\partial A_\theta}{\partial r} \frac{\partial r_\theta}{\partial V_\theta} = \frac{2}{r}$$
 Eq. (5)


179 By inserting Eq. 5 b) into Eq. 4 we obtain our final expressions describing the change in free

180 energy of soil water as function of a change in the stored water in the control volume:


$$a) \; dE_{free} = \rho g z_{GW} dV_\theta + \rho g \psi(\theta) dV_\theta \Leftrightarrow$$
$$b) \; \frac{\partial E_{free}}{\partial V_\theta} = \rho g z_{GW} + \rho g \psi(\theta)$$
 Eq. (6)


184 In the following we denote the first term on the right hand side as potential energy and the

185 second one as capillary surface energy of soil water. Note that the latter is negative as the soil

186 water potential is as a suction head negative as well. The stored water amount in a small

187 control volume is equal to the product of the volume V and of the soil water content $\theta$ ($m^3 m^{-3}$

188 ). Hence, a change in the stored water amount relates either to a dynamic change in the soil

189 water content, while the control volume size remains constant, or an increasing size of the

190 control volume when moving up scale at a constant time:


$$V_\theta = \theta V \Leftrightarrow$$
$$dV_\theta = V d\theta + \theta dV$$
 Eq. (7)


194 Local dynamic changes in the soil water stock, usually described by the Darcy-Richards

195 equation, change thus the local free energy state of the soil water as well:


$$\frac{1}{V} \frac{\partial E_{free}}{\partial t} = \rho g z_{GW} \frac{\partial \theta}{\partial t} + \rho g \frac{\partial \psi(\theta)}{\partial \theta} \frac{\partial \theta}{\partial t}$$
 Eq. (8)


199 This opens opportunities to analyze and visualize soil water dynamics through changes of the

200 corresponding free energy state, as further detailed below. From equations 6 and 7 we can





also derive the free energy of soil water stored in a finite control volume at a constant time.
This is in fact equal to the integral of the product of the total hydraulic potential, $\psi + z_{GW,}$ and
the soil water content over the volume of interest (de Rooij, 2009; Zehe et al., 2013):

$$E_{free} = E_{cap} + E_{pot} = \int \rho g (\psi(\theta) + z_{GW}) \theta dV \quad \text{Eq. (9)}$$

The two drivers in Darcy's law, the soil water potential and the gravity potential, reflect thus
in fact the weight-specific capillary surface energy and the weight specific potential energy of
soil water. Note that the potential energy of soil water grows with increasing storage while
capillary surface energy shrinks as the soil water potential declines with increasing wetness.
From equation 8 it becomes furthermore clear that capillary surface energy is in accordance
with the non-linear shape of the soil water retention curve the main source of non-linearity in
soil water dynamics and in its free energy state, because it scales with the slope of the
retention curve. The energy perspective reveals, however, nicely that potential energy of soil
water is at a given elevation above the groundwater surface a linear function of the soil water
content. This already indicates that dominance of the one or the other energy form is
important for the question whether a system behaves in a linear or non-linear fashion.
## 2.2  Equilibrium storage and energy state at a depth to groundwater
The state of minimum free energy is reached when $\partial E_{free}/\partial V_\theta == 0$. Due to Eq. (6) this is the
case when the system is in hydraulic equilibrium, where $\Psi$ equals the negative of $z_{GW}$
everywhere in the subsurface:

$$\rho g (\psi + z_{GW})\theta = 0 \Leftrightarrow \\ \psi = -z_{GW} \quad \text{(Eq. 10)}$$

The soil hydraulic equilibrium corresponds hence to a state where the absolute value of the
free energy of soil water is minimal, because the specific potential energy of soil water equals
its specific capillary surface energy density at any point in the subsurface. Note that this
means equivalently that the system is in a state of perfect mixing, and thus maximum mixing
entropy, due to the absent gradient in hydraulic potential (Kondepudi and Prigogine, 1998;
Iwata et al., 1995). The equilibrium storage at any point in the system can be inferred from the




water retention curves of the soils in a straightforward manner by substituting the soil water
potential by the depth to the groundwater water surface (e.g. Porada et al., 2011):

$$S_{eq} \equiv \frac{\theta}{\theta_s} \big|(\psi = -z_{GW})  \text{ Eq. (11)}$$


Where $\theta_s$ ($m^3 m^{-3}$) is the saturated soil water content and S (-) is the relative saturation. This is
illustrated in figure 2 for the retention curves of three distinctly different soils, assuming
arbitrarily a depth to groundwater of $z_{GW}$ = 10 m. The equilibrium saturation of the clay rich
soil in the Marl geological setting of the Wollefsbach catchment is with $S_{eq}$= 0.82 rather large,
while the young silty soil located in the Colpach has a rather small saturation at equilibrium of
$S_{eq}$=0.13. The loess soil from the Weiherbach is with $S_{eq}$= 0.53 in between these extremes.
Note that two of those soils are located in our respective study areas Colpach and Wollefsbach
(compare section 3). We added the Weiherbach soil to complete the spectrum of possible
endmembers.

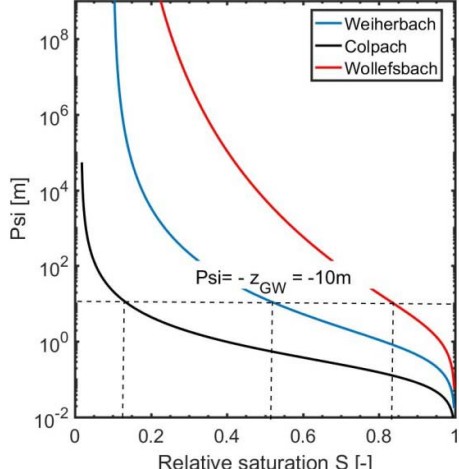

Figure 2: Soil water retention curves as function of relative saturation determined as explained in
section 3.3. The dashed black lines mark the relative saturation at hydraulic equilibrium, assuming
arbitrarily a depth to groundwater of $z_{GW}$ = 10 m. The Wollefsbach and the Colpach are further
characterised in section 3.





Note that although these values are very different in magnitude, they represent the respective
equilibrium storage states, which these systems at this elevation will naturally approach when
relaxing from external disturbances. And it is exactly those equilibrium storages which
separate the aforementioned state ranges where the system is either in a storage deficit or in a
storage excess. This becomes obvious when plotting the specific free energy per unit volume
$e_{free}$ (m) of the soil water stock as function of the relative saturation for these soils (Fig. 3).
The latter can be obtained by deriving Eq. 9 with respect to V and normalising it with $\rho$ g:

$$e_{free} \equiv \frac{1}{\rho g} \frac{\partial E_{free}}{\partial V} \equiv \left( \psi(\theta) + z_{GW} \right) \cdot \theta = f\left( \frac{\theta}{\theta s} \Big|_{z_{GW}} = const \right) \text{ Eq. (12)}$$


Note $e_{free}$ is, as being defined as free energy per unit volume, equal to the product of total
hydraulic potential and the soil water content. It thus differs from the total hydraulic potential,
which is the free energy per wetted control volume.

The horizontal green line in Figure 3 marks the local equilibrium state where the absolute
value of the specific free energy at this particular elevation is zero. The vertical lines indicate
the corresponding equilibrium saturations at the x-axis (these correspond to those in figure 2).
In case of a constant depth to groundwater, these equilibrium storages separate the ranges of
soil saturation belonging to the P-regime (in blue) from those that belong to the C-regime (in
red), respectively. In the P-regime $e_{free}$ is positive as $e_{pot}$ is larger than the absolute value of
$e_{cap}$. Storage dynamics are hence dominated by potential energy differences and thus gravity.
In the P-regime, the system is locally in a state of storage excess because it has with respect to
its equilibrium too much potential energy. One might thus expect that the system needs - in
absence of an external rainfall forcing - to release water from this elevation to relax back to
equilibrium. Note that the necessary amount of water that needs to be released is determined
by the overshoot of free energy above zero. In the C-regime specific free energy gets negative
as capillary surface energy becomes larger than the potential energy. Storage dynamics are
dominated by local capillary controls. The system needs to recharge water to deplete the "free
energy deficit" below zero, and the necessary amount depends on the free energy deficit
below zero.




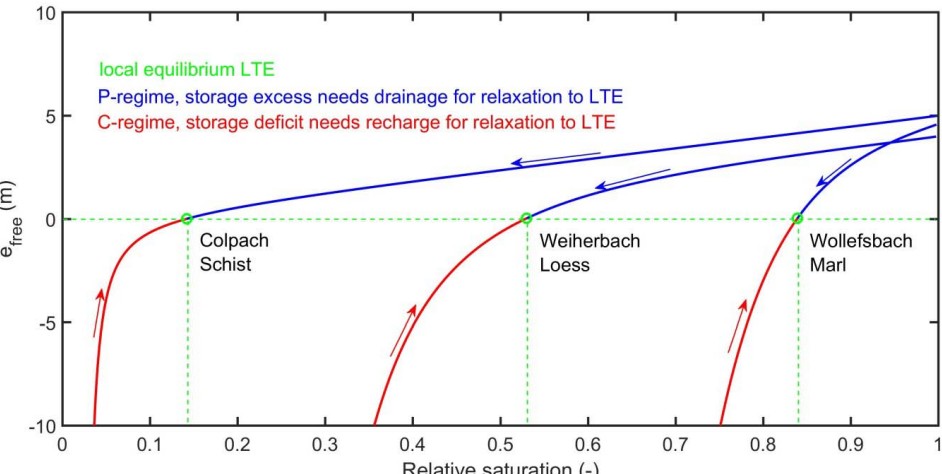


Figure 3: Weight specific free energy state of the soil water stock, as defined in Eq. (12), plotted
against the relative saturation of the three different soils, assuming a depth to groundwater of 10m.
The green lines mark the local equilibrium state where the absolute value of the specific free energy is
zero and the corresponding equilibrium saturations. Free energy in the P-regime and C-regimes are
plotted in solid blue and red respectively, the arrows indicate the way back to equilibrium.

Note that we assume the local soil volume to be in capillary contact with the groundwater
surface (see section 5.2 for further discussion). It is clear that in case of a dynamic
groundwater table, the equilibrium storage and the energy state function at a distinct depth
will change. The same holds in case of a constant depth to groundwater, when moving
vertically through the unsaturated zone, as explained in the next section.

## 2.3   The energy state function of the unsaturated zone

From equation 12 and the graphs in Fig. 3 it is only a small step to derive the energy state
function for the unsaturated zone of any hydrological system, with a fixed groundwater level
or where the groundwater level is known as function of time. Specific free energy of the soil
water stock depends jointly on the retention curve of a point in the unsaturated zone and its
elevation above the groundwater surface. This implies that the energy state curves in a system
of interest change also with the range of elevations above groundwater. In the following we
illustrate this for the idealized case of a landscape with a single soil and a well-defined single
retention function.





Common ways to characterize the topographic distribution in a catchment are either by means
of the hypsometric integral, taking the catchment outlet as reference, or by means of the
height over nearest drainage (HAND, Renno et al., 2008; Nobre et al., 2011), taking the
closest stream as reference. Here we extend the idea of HAND from the land surface to the
entire unsaturated zone of a catchment, using optionally the water level in the next channel as
a proxy for the depth of the groundwater surface and its changes. We may hence characterize
the 'family' of energy state curves describing free energy of soil water at different elevations
above groundwater if we know a) the retention functions of the soils and b) the range of
HAND, $R(z_{GW})$, in the system of interest, as follows:

$$e_{free}(\text{soil}, z_{GW}) = \left(\psi_{soiltype}(\theta) + z_{GW}\right) \cdot \theta = f\left(\frac{\theta}{\theta s}, z_{GW}\right), z_{GW} \in R(z_{GW}) \quad \text{Eq. (13)}$$

The energy state function consists thus of a family of curves, characterizing how depth to
groundwater and soil physical characteristics jointly control the free energy state of soil water
as function of the relative saturation. Please note that all points with the same soil water
retention curve and the same elevation above groundwater are represented by the same energy
state curve.

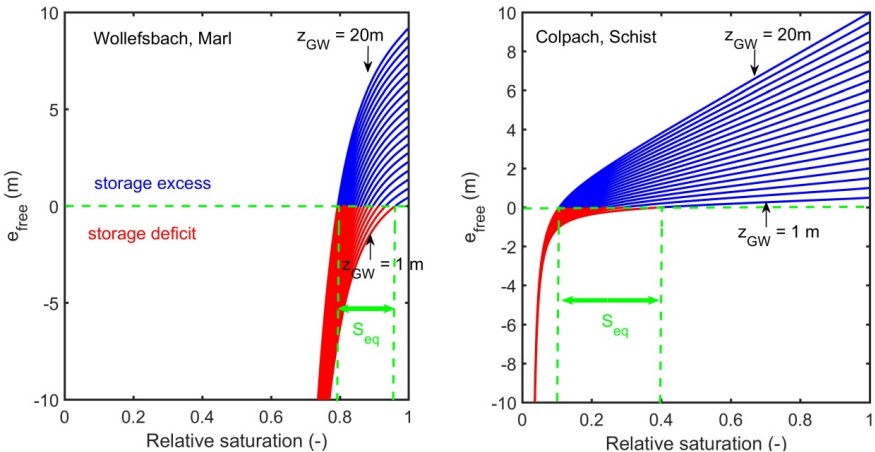


Figure 4: Energy state function, characterizing the specific free energy state of soil water as function
or relative saturation, soil water retention and depth to groundwater. Each of the 20 curves represents a
distinct depth above the groundwater surface for a discrete range of 1m to 20 m. The vertical green
lines mark the range of the equilibrium saturations $S_{eq}$ at the different elevations.





Figure 4 gives an impression of how the energy state functions would look like, in case the
soils of the Colpach and the Wollefsbach were distributed along an elevation range above
groundwater of R=[1m, 20m]. Note that we assume that the soil water retention function in
figure 1 is valid everywhere in this hypothetical landscape. Figure 4 depicts that the individual
energy state curves of the family become generally steeper and the P-range becomes generally
wider with an increasing depth to groundwater. This reflects a) the increasing importance of
potential energy and b) the decreasing equilibrium saturation with increasing depth to
groundwater. The shape of the individual curves and the equilibrium storage are for both
hypothetical systems distinctly different. Given those strongly different shapes of the energy
state functions one might expect the two systems to exhibit strongly different storage
dynamics. We further elaborate on these differences in section 3, when introducing the energy
state functions of our study areas.

## 3    APPLICATION

The energy state function introduced in the last section defines the space of possible energy
states of the soil water stock, a thermodynamic state space of the unsaturated zone at the level
system so to say. Due to the intermittent atmospheric forcing and the exchange of the soil
water with atmosphere, groundwater body and river, parts of the system are frequently pushed
and pulled out of its equilibrium into states of the either P or the C regime. It appears thus
straightforward to visualize these storage dynamics, either observed or modeled, as pseudo[2]
oscillations of the corresponding free energy state in the respective energy state functions.
This will teach us a) which part of the state space is actually visited by the system, and b)
whether the system predominantly operates in one of these regimes or within both them. In
the following we briefly characterize the study areas and the dataset we use for this purpose.

### 3.1    Study areas and their energy state functions

The Colpach and the Wollefsbach catchments belong to the Attert experimental basin (Pfister
et al., 2002; Pfister et al., 2017), and are distinctly different with respects to soils, topography,
geology and landuse (Fig. 5a). Both catchments have been extensively characterized in
previous studies with respect to their physiographic characteristics, dominant runoff
generation mechanisms and available data (Wrede et al., 2015; Martinez-Carreras et al., 2015;
Loritz et al., 2017; Angermann et al., 2017). Hence, we focus here exclusively on those

---

[2] We use the term pseudo here, as these are in fact deviations into different directions and relaxations to LTE. These are no oscillations in a strictly mechanical sense.





system characteristics which determine their respective energy state functions. The Colpach
has an elevation range from 265 to 512 m. Soils are young silty haplic Cambisols that formed
on schistose periglacial deposits. Despite of their high silt content they are characterized by a
high permeability and high porosity (Jackisch et al., 2017), because the fine silt aggregates
embed a fast draining network of coarse inter-aggregate pores. In contrary, the Wollefsbach
has a much more gentle topography from 245 to 306 to m.a.s.l. Soils in this marl geological
setting range from sandy loams to thick clay lenses. Soil water retention was in both
catchments analyzed by Jackisch (2015) using a set of 62 undisturbed soil cores from the
Colpach and 28 undisturbed soil cores from the Wollefsbach.
Here we do not use these point relations but a representative, macroscale soil water retention
function to derive the energy state function of our study areas. These were derived by
Jackisch (2015) from the raw data of all experiments as follows. He pooled the matching pairs
of soil water content and matric potential of all experiments in a landscape into a single
sample (Fig. 5 b and c), which hence characterizes the spreading of saturations at a given
tension that occurred in these systems. By averaging the soil water content at each matric
potential/tension-level he obtained an aggregated data set characterizing the relation between
the average soil water content that is stored at a given soil water potential/tension. The
representative retention curves were finally optioned by fitting the van Genuchten-Mualem
relation to the aggregate data (Jackisch, 2015). Note that this relation cannot be observed at a
single site, it is a macroscale relation which characterizes the average behavior in the entire
system. More importantly this approach preserves the relation between the average soil water
content and the specific capillary surface energy and it has been shown to perform superior
during a process based simulation of the water balance in both areas, which represented the
system by a single representative hillslope (Loritz et al, 2017). The topography of these
representative hillslopes corresponds in both cases to the distribution of HAND, which
implies that the distribution function of potential energy along the flow path to the stream is
preserved (Fig. 5d). By combining these representative hillslopes with the representative
retention functions we finally obtained the energy state functions of the Colpach (Fig. 6 a) and
the Wollefsbach (Fig. 6 b).



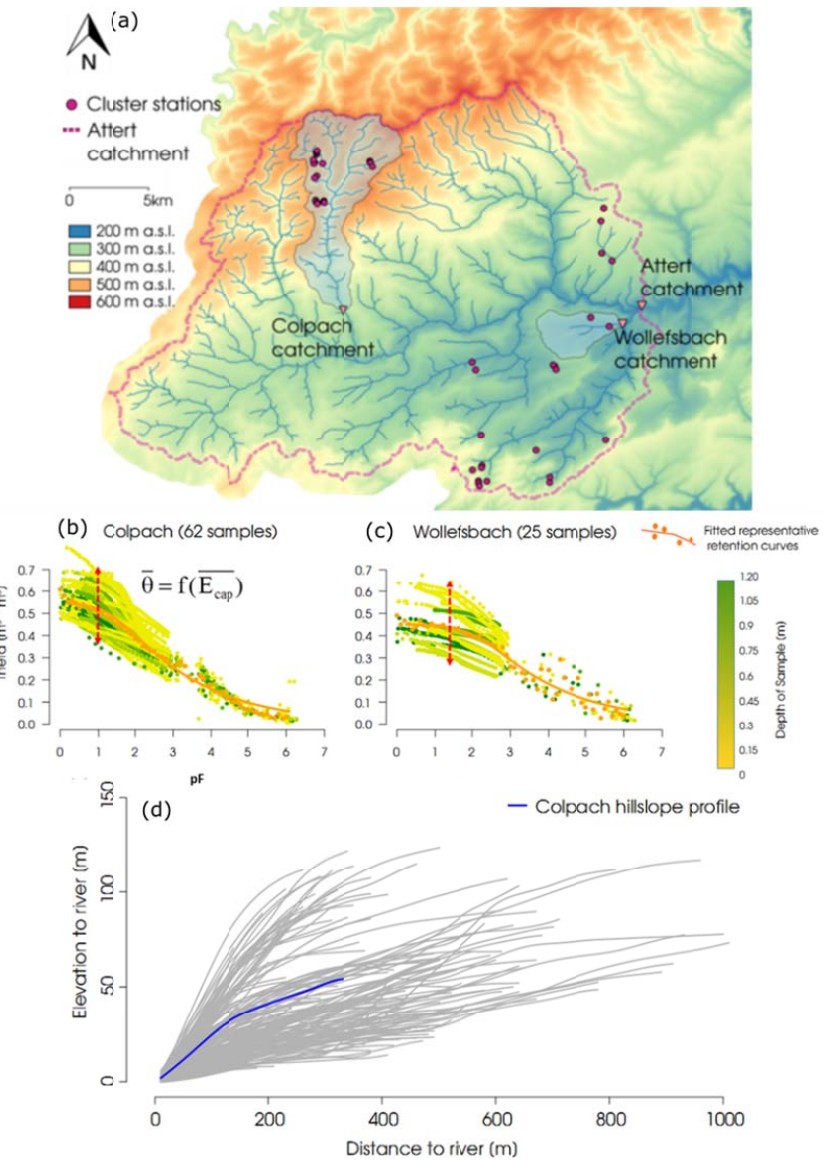


Figure 5: Map of the Attert basin with the Colpach and Wollefsbach catchments (Panel a, taken from
Loritz et al. 2017). The red dots mark the cluster sites of the CAOS research unit, which collect
besides the standard hydro-meteorological data, soil moisture and the soil water potential. Panel b and
c present the data from the soil water content as function of tension observed in a large set of multistep
outflow experiments and the representative retention curves obtained by energy conservative
averaging. Panel d show the distribution of HAND for the hillslope in the Colpach catchment and of
the effective hillslope Loritz et al. (2017) derived for his model study, the corresponding distribution
in the Wollefsbach is not shown, as it is due to the homogeneous topography rather straightforward.



Note that the larger elevation range in the Colpach causes a clear dominance of the P-Regime
over a wide range of saturation. More importantly figure 6a reveals that for relative
saturations larger than 0.4 free energy is a multilinear function of relative saturation. This
means that the specific free energy is at each geopotential level a linear function of relative
saturation, but the slope of the energy state curves does increase with increasing distance to
groundwater. In contrary the Wollefsbach is clearly a non-linear system within the entire
range of elevations, with clearly much smaller maximum free energy but with a huge potential
for strongly negative free energy when soils dry out. Consistently, we find the ranges of
equilibrium saturation in the systems to be rather different, between 0.95 and 0.78 in the
Wollefsbach and between 0.4 down to 0.18 in the Colpach.

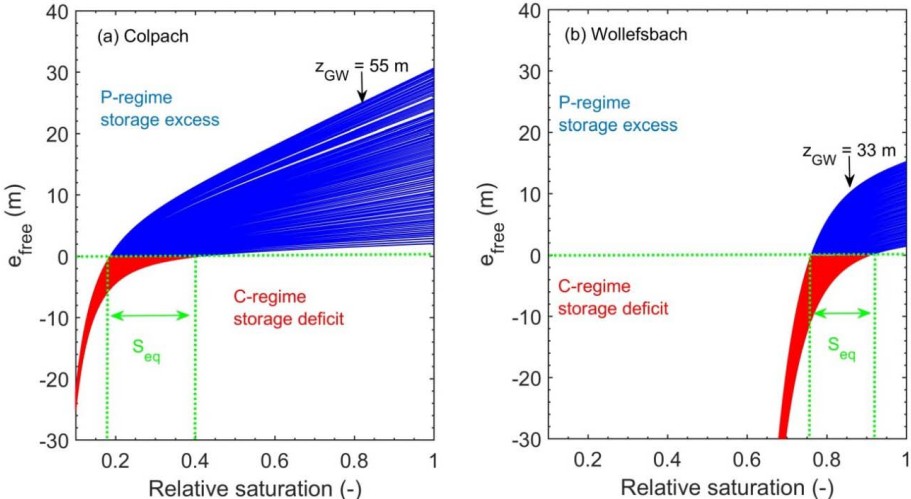


Figure 6: Energy state functions of the Colpach (a) and the Wollefsbach (b) derived from the range of
HAND and the representative retention functions in figure 4. The horizontal green line mark the
equilibrium of zero free energy, the vertical green lines mark the corresponding ranges of equilibrium
saturations.

### 3.2    Available storage observations

We use data from a distributed network of 45 sensor clusters spread across the entire Attert
experimental basin (Figure 4) collected within the hydrological year 2013/14. These clusters
measure, among other variables, soil moisture and soil water potentials within three replicated
profiles in 0.1, 0.3 and 0.5 m depths using Decagon 5TE capacitive soil moisture sensors. As





direct observations of groundwater levels are rare and only available close to riparian zone we
use the height over the next stream to estimate $z_{GW}$.

## 4    RESULTS

### 4.1    Soil moisture and its free energy state at two distinct cluster sites

In a first step we inter-compare the free energy states of the soil moisture stock (Fig. 7) which
was observed at two arbitrarily selected sites in the respective study catchments. Both sites are
located 20 m above their respective streams. The soil water content in the clay rich top soil of
the Wollefsbach site is in the winter and fall period rather uniform and on average 0.12 $m^3m^{-3}$
larger than in the Colpach (Fig. 7a). While the soil water content at the Colpach site appears
much more variable in these periods. Both sites dry out considerably during the summer
period and start to recharge with the beginning of the fall.

Figure 7 b and c provide the corresponding free energy states of both soil water time series as
function of the soil saturation. Observations are shown as black circles and the related
theoretical energy state curves, calculated after Eq. 12. The first thing to note is that the
observed free energy states for both sites scatter nicely around the theoretical curves. More
interestingly one can see that the spreading of the free energy state of the soil water stock is at
both sites distinctly different, while the ranges of the corresponding soil water contents are
comparable. The free energy state of soil water at the Colpach site is during the entire
hydrological year in the P-regime and hence subject to an overshoot in potential energy (Fig.
7b). The site operates in the linear range of the energy state curve and fluctuates around an
average energy height of 6.0 m, which corresponds to an average energy density of $5.9*10^4$
$Jm^{-3}$. While the observations spread across a total range of 3 m ($2.9\ 10^4\ Jm^{-3}$) their standard
deviation is 0.31 m ($3.0*10^3\ Jm^{-3}$). The coefficient of variation of the free energy state is
hence with 0.05 rather small. In contrary the specific free energy of the soil water stock in the
Wollefsbach spreads across a much wider range of almost 50m, which corresponds to $4.9*10^5$
$Jm^{-3}$ (Fig. 7c). The average specific free energy is with 4.8 m ($4.7*10^4\ Jm^{-3}$) clearly smaller as
at the Colpach site, while the coefficient of variation is with 1.3 much larger. Most
importantly the system operates qualitatively differently as it switches to the C regime and



thus a strong storage deficit during dry spells in the summer period. Please note that the free
energy declines to the value corresponding to the permanent wilting point pwp[3].

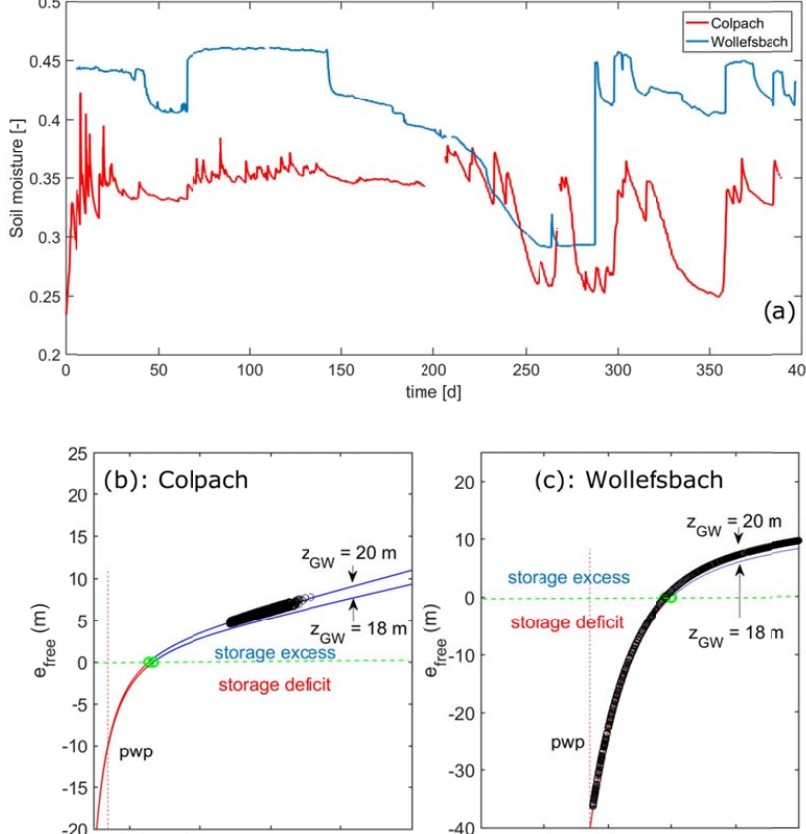

Figure 7: Top soil water content observed at cluster sites in the Colpach and the Wollefsbach
catchment (panel a) and the corresponding free energy states in their respective energy state curves
(panel b and c). The black circles mark the observations. The vertical dashed line marks the permanent
wilting point, which is due to the definition of the total free energy in Eq. 6 not simply equal to a total
hydraulic potential of -133 m. Panels b and c show additionally the energy state curve for $z_{GW}$ = 18 m,
to highlight that an error in the estimated depth to groundwater implies a substantial mismatch
between observations and the theoretically predicted curve.

---

[3] As specific free energy is according to Eq. (12) the product of the total soil hydraulic potential and the soil water content, its value at the pwp does not simply correspond to – 133m.



We hence state that the free energy state of the soil water stock reveals a distinctly different
dynamic behavior of both sites, which cannot be derived from the inter comparison of the
corresponding soil water moisture time series. The Colpach site is characterized by permanent
storage excess, though the corresponding soil water content is nearly always smaller than in
the Wollefsbach. Free energy of the soil water stock is a linear function of relative saturation.
In contrary, the Wollefsbach shows a strongly non-linear behavior at this site and it switches
to a storage deficit when the soil saturation drops below 0.78, which corresponds to a soil
water content of 0.374 $m^3m^{-3}$ (Fig. 7a). We thus wonder whether the term wet and dry should
therefore be used with respect to the equilibrium storage as meaningful reference point. Last
but not least the theoretical energy level curves derived for a distance to groundwater of $z_{GW}$ =
18 m do not fit the corresponding observations but show a clear negative bias (Figure 7b,c).
An error in the estimated depth to groundwater implies thus a substantial mismatch between
the observations and the theoretically predicted energy curve. This implies that energy levels
will also change with changing groundwater surface, as further detailed in the discussion.

### 4.2   Soil moisture and its free energy state within the entire observation domain

Figure 8 presents the free energy states of the top soil moisture which was observed at all
cluster sites in the Colpach (panel a, N = 41) and the Wollefsbach (panel b, N = 20). The
respective heights above the channel range from 1 to 45 m in the Colpach and from 1 to 22m
in the Wollefsbach. Generally, the observed free energy states scatter again nicely around the
energy state curves of the corresponding $z_{GW}$. The Colpach operates, except for the sites
located at the smallest distances to groundwater ($z_{GW}$ =1 and 4m), in the linear range of the P-
regime, indicating that soil moisture dynamics is dominated by potential energy differences.
Free energy of the soil water stock is hence a multi linear function of saturation. The total set
of observations in the Colpach generally spread across a wide range of relative saturations,
and the corresponding "amplitudes" of the free energy deviations are clearly larger as at the
single site shown in Figure 7 b. This is because sensor clusters with the same estimated height
above groundwater were pooled into the same subsample regardless of their separating
distance. For instance at $z_{GW}$ = 1 m the subsample consisted of 1 cluster with three replicate
soil moisture profiles, at $z_{GW}$ = 17 we had for instance 3 sensor clusters and thus in total 9 soil
moisture profiles. The partly large spreading of the observations may hence be explained by a
combination of local scale heterogeneity and large scale differences in the drivers of soil
water dynamics such as rainfall or local characteristics of forest vegetation.



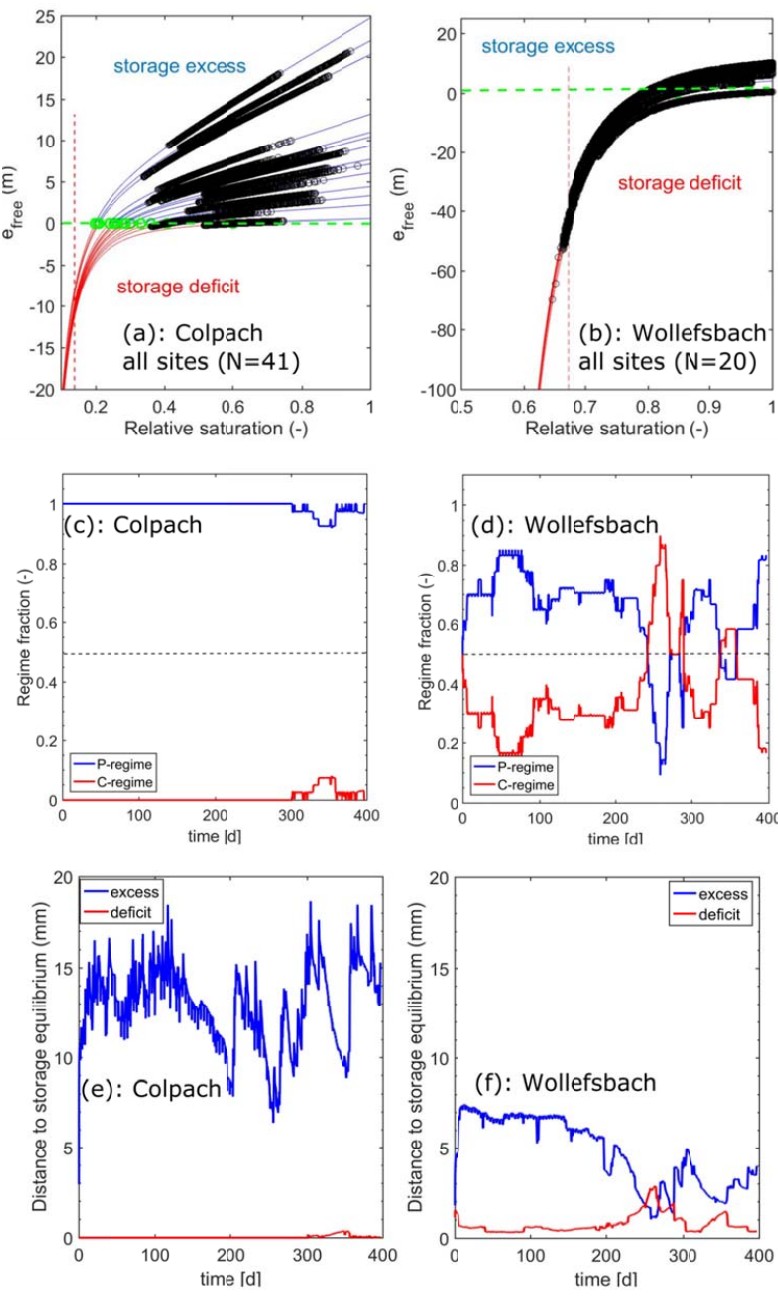


Figure 8: Free energy of all observations in the Colpach (a) and Wollefsbach (b) plotted in their
corresponding energy state function (note the different scales). The black circles mark the
observations. The horizontal green lines mark the equilibrium of zero free energy. Panel c and d show
which fractions of the data set was in the P or in the C regime as function of time. Panel e and f show
the averaged distance to storage equilibrium for the sites in a storage deficit and a storage excess.



Despite of the large spreading between 80 -100% of the Colpach sites operated permanently
in the P-Regime (Fig. 8c). During the wet season it is 100% of the sites, between day 300 and
400, the profiles at the lowest distance to groundwater switch into the C-regime and thus a
storage deficit. Fig. 8 e shows the average distance to storage equilibrium in mm as function
of time. Sites with storage excess are on average 13 mm above their equilibrium, the
minimum is 6 mm. The average top soil storage deficit (we show this as positive value) is the
entire observation period rather small. This suggests that the top soil sites in the Colpach may
the during entire observation period release water either to the subsoil or in down slope
direction.

In the Wollefsbach we find, consistently with figure 7b, a drop of free energy into the C-
regime during the dry summer period. Storage dynamics in the entire system works, in
accordance with the shape of the energy level function, in a strongly non-linear manner. Note
that a few values drop even below the permanent wilting point. This can either be explained
by the fact that local vegetation is more efficient in root water uptake as sun flowers (the
model crop to define the permanent wilting point), or by measurement errors. In contrary to
the Colpach, the fractions of profiles which operate in the C-regime or in the P-regime are
much more variable in time (Fig. 8d). On average 30% of the profiles operate in a storage
deficit during the entire observation period, the minimum is 15% and the C-regime fraction
peaks at 90% at day 250. Consistently, we find that the storage excess in the Wollefsbach is
on average 3 mm, while the average storage deficit is much more prominent; it even exceeds
the storage excess during the summer period (Fig. 8e). These differences are consistent with
the strongly different runoff generation behavior of both systems, as further detailed in the
discussion.
## 5   DISCUSSION AND CONCLUSIONS
The presented results provide strong evidence that a thermodynamic perspective on soil water
storage provides holistic information for judging and inter-comparing the storage state and
storage dynamics of hydrological systems, which cannot be inferred from soil moisture
observations alone. In the following we reflect the general idea of using free energy as state
measure, discuss its promises as well as its limiting assumptions. We then move on to the
more specific differences in the storage dynamics in both our study systems. And we close by
reflecting on the seeming paradox between the known local non-linearity of soil physical




characteristics and the frequent argumentation that hydrological systems often behave much
more linearly.

### 5.1    Free energy and the energy level function – options and limitations

Our results clearly show that free energy as function of relative soil saturation holds the key to
defining a meaningful state space of a hydrological system, regardless of its spatial extent.
This space of possible energy states consists of a family of energy state curves, where each of
those characterizes how free energy density evolves at a distinct elevation above ground
water, depending on the triad of the matric potential, gravity potential (i.e. depth to
groundwater) and soil water content. The free energy state of soil water reflects in fact the
balance between its capillary surface energy and geo potential energy densities and we
showed that this balance determines:
•   Whether a system is at given elevation above groundwater locally in its equilibrium

storage state ($e_{free} == 0$), in a state of a storage deficit ($e_{free} < 0$) or in state of a storage

excess ($e_{free} > 0$);

•   The regime of storage dynamics. Soil water dynamics in the C-regime ($e_{free} < 0$) are

dominated by capillarity i.e. the local, non-linear soil physical driver, which means the

system needs recharge to relax to its equilibrium. Or it is in the P-regime ($e_{free} > 0$)

dominated by potential energy, i.e. the non-local linear gravitational control, which

means the system needs to release water to relax to local equilibrium.

The energy level function turned out to be useful for inter-comparing distributed soil moisture
observations among different hydrological landscapes, as it shows the trajectory of single sites
or of the complete set of observations in its state space. This teaches us which part of the state
space is actually 'visited' by the system during the course of time, whether the system
operates predominantly in a single regime, whether it switches between both regimes and how
much water needs to be released or recharged locally for relaxing back to local equilibrium
and how often it actually is at equilibrium or if it never gets there. Note that the usual
comparison of soil water contents alone did not yield this information. On the contrary from
this we would conclude that the site in the Wollefsbach is, due to the higher soil water
content, always 'wetter' than the corresponding site in the Colpach. The free energy state
reveals, however, the exact opposite, we have a storage excess at Colpach site for the entire
year while the Wollefsbach site is in summer in a storage deficit. We thus propose that the



term wet and dry should only be used with respect to the equilibrium storage as meaningful
reference point.

The apparent strong sensitivity of the free energy state of the soil water stock to the estimated
depth to groundwater offers on the one hand new opportunities for data based learning and an
improved design of measurement campaigns, but it also determines the limit of the proposed
approach. With respect to the first aspect, we could show that an error of 2 m in the assumed
depth to groundwater lead to a clear deviation of the observed free energy states from the
theoretical energy level curve. This offers either the opportunity to estimate depth to
groundwater from joint observations of soil moisture and matric potential, in case the local
retention function is known. This can be done by minimizing the residuals between the
observation and the theoretical curve as function of depth to groundwater. Or it allows for the
derivation of a retention function based on the joint observations of soil moisture, matric
potential and depth to groundwater. Here, we need again to minimize the residuals between
the observation and the theoretical curve but this time as function of the parameters of the soil
water retention curve. Due this strong sensitivity it is furthermore important to stratify soil
moisture observations both according to the installed depth of the probe and according to the
elevation of the site above groundwater, or the height over the next stream. The former is
important because the soil above overlaying the sensor acts as a low pass filter. The latter is
important because depth to groundwater determines the equilibrium storage the site will
approach when relaxing from external forcing.

Despite of all these opportunities for learning, the sensitivity of free energy to the depth to
groundwater implies that the site of the system is still in hydraulic contact with the aquifer.
This key assumption is certainly violated if the soil gets so dry that the water phase becomes
immobile while the air phase becomes the mobile phase. And it might get violated if depth to
groundwater becomes too large. Last but not least the groundwater surface may change either
seasonally, or in some systems more rapidly, and this changes $z_{GW}(t)$ in the energy level
function and the storage equilibrium. We nevertheless conclude that it is worth to collect joint
data sets either of the triple of soil moisture, matric potential and the retention function at
distributed locations (as we did in the CAOS research unit as explained in (Zehe et al. 2014))
or even preferable on the quadruple of soil moisture, matric potential, retention function and





depth to groundwater. Soil moisture observations alone appear not very informative about the
system state.

## 5.2 Storage dynamics in different landscapes – local versus non local controls

More specifically we found that soil water dynamics in the Colpach and the Wollefsbach
exhibit a substantial difference. The observations clearly revealed that the Colpach operates
the entire hydrological year in a state of storage excess due to an overshoot in potential
energy. Soil water dynamics is mainly driven by potential energy, which means that the linear
and non-local gravitational control is dominant. Most interestingly we found that the free
energy state of the soil operated in the linear range of the P-regime, which implies that the
storage dynamics is (multi) linear. This means that the specific free energy is at each
geopotential level a linear function of relative saturation, but the slope of the energy state
curves does increase with increasing distance to groundwater

We found furthermore that the annual variation of the averaged free energy of the soil water
stock was rather small. Zehe et al. (2013) found a similar, almost steady state behavior, for the
free energy of the soil water stock in the Mallalcahuello catchment in Chile, which also
operated in the P-regime the entire year. Note that both landscapes are characterized by a
pronounced topography, by well drained highly porous soils (Blume et al., 2008a;Blume et
al., 2008b;Blume et al., 2009) and that both are predominantly forested. And in both
landscape subsurface storm flow is the dominant runoff generation process, as gravity is the
dominant control of soil water dynamics.
On the contrary the Wollefsbach was characterized by a seasonal change between both
regimes: operation in the P-regime during the wet season and a drop to a strong storage deficit
during the dry summer period when operation in the C-regime. Free energy was at all sites is
a clearly non-linear function of the relative saturation. Interestingly we found the same
seasonality for the Weiherbach catchment in Germany, a dominance of potential energy
during the wet season and a strong dominance of capillary surface energy in summer (Zehe et
al 2013). Note that both landscapes are characterized by silty cohesive soils and a gentle
topography and both are used for agriculture. In both areas Hortonian overland flow would
play the dominant role, but this process is actually strongly reduced due to a large amount of
worm burrows acting as macropores. Both landscape are also controlled by tile drains, which
artificially controls depth to groundwater. In both areas the soil water dynamics is dominated



by capillarity during the summer period, which means that the local soil physical control
dominates also at the system level.

We thus suggest that similarity in those landscape attributes - which controls the energy level
function - implies qualitatively identical regimes of storage dynamics. We furthermore
wonder whether distinct differences in the energy state functions and more importantly of the
free energy states might help explaining differences in dominant runoff generation. In the
Colpach we found a clear storage excess in the top soil during the entire observation period,
while the in Wollefsbach the storage deficit exceeded storage excess in summer. This
difference might explain the strong difference in runoff generation. The Colpach is
characterized by a strong subsurface and base flow component while those are negligible in
the Wollefsbach.

### 5.3   Free energy to assist hydrological predictions

Although the scope of this study was on the usefulness of thermodynamics to diagnose and
explain different storage dynamics, we will add a short discussion on the predictive value of
this thermodynamic perspective. In this context it is important to recall that relaxation back to
equilibrium and thus dissipation of free energy is in both regimes accelerated by different
types of preferential pathways. Zehe et al (2013) distinguished wetting structure which favor
recharge of the dry soil matrix and deplete the storage deficit, from drainage structures which
favor water release and deplete the storage excess, because these affect free energy dynamics
of the soil water stock in largely different ways.

Drainage structures are preferential pathways that extend continuously through the
unsaturated zone either into the aquifer or to the riparian zone. Typical examples are
macropores that connect to subsurface pipe networks or tile drains (Zhang et al., 2006;Weiler
and McDonnell, 2007;Wienhofer et al., 2009;Wienhofer and Zehe, 2014;Nimmo, 2012,
2016;Gelbrecht et al., 2005;Klaus and Zehe, 2011;Klaus et al., 2014). They facilitate
bypassing and export of excess water and hence act similar to veins. They thereby accelerate
reduction of potential energy overshoot and thus relaxation from storage excess in the P-
regime. For systems which operate predominantly in the P-regime it is important to recall that
a steady state in the free energy balance does not imply a steady state of the water balance.
This has been shown by Zehe et al. (2013), by analyzing the free energy balance of rainfall
input, soil water storage and runoff. It does however imply a constant ratio between steady





storage and release, which can be calculated based on the upslope geo-potential and the
elevation where the system exports runoff to the stream (Zehe et al. 2013). For the
Mallacahuello catchment (Zehe et al. 2013) this turned out to be a reasonable estimate of the
average annual runoff coefficient. And a model structure which was, with respect to the
density of drainage structures, tuned to reproduce this energetic steady state was shown to
provide acceptable simulations of stream flow. It might thus valuable to test whether we find
similar results for the Colpach.

The relaxation from the C-regime requires macropores which facilitate wetting and recharge
of the subsoil from dry initial conditions. This implies that these "wetting structures" do not
extend across the entire unsaturated zone, but end within the subsoil; typical examples are
worm burrows (Shipitalo and Edwards, 1996; Shipitalo and Butt, 1999; Zehe and Fluhler,
2001; Bastardie et al., 2003; Lindenmaier et al., 2005; Binet et al., 2006; van Schaik et al.,
2014) or shrinkage cracks (Vogel et al., 2005a; Vogel et al., 2005b; Zehe et al., 2007) or root
channels (Tobón-Marin et al., 2000; Abernethy and Rutherfurd, 2001; Gregory, 2006;
Johnson and Lehmann, 2006; Tietjen et al., 2009) or a network of conductive inter-aggregate
pores (Jackisch 2015; Jackisch et al. 2017). Wetting structures favor recharge as they allow a
bypassing of the dry and thus low conductive soil matrix and a subsequent wetting of the
subsoil across the macropore matric interface (Beven and Germann, 2013; Jackisch and Zehe,
2018). They thereby accelerate depletion of gradients in soil water potential, reduce capillary
binding energy excess and accelerate relaxation from the storage deficit in the C-regime. In
those systems, which seasonally switch among both regimes, we may find a thermodynamic
optimal density of wetting structures that maximizes average dissipation of free energy during
recharge events. Zehe et al. (2013) showed that this optimum structure, which balances
recharge and surface runoff in an optimal way, allowed successful predictions of rainfall
runoff generation and the water balance in the Weiherbach catchment.

We thus conclude that idea of an optimum catchment configuration postulated by Savenije
and Hrachowitz (2017) is supported by our study. It might correspond to a thermodynamic
optimum and exist in those systems, which seasonally switch between the P and the C regime.
Note that such thermodynamic optimum maximizes dissipation of free energy during
relaxation back to equilibrium (Zehe et al. 2013), and this implies that the relaxation time
becomes minimal. Zehe et al. (2013) showed that such an optimum requires a preferential



flow network that balances wetting and drainage. Along similar lines the free energy state of
simulations might help in model assessment and evaluation. This approach is promising to
discriminate substantial model errors from insubstantial ones. A substantial error could be
defined for instance when the model and observation operate for longer periods in different
energetic regimes. Alternatively, one can explore how changes in system characteristics,
particularly macropores, affect how the system operates in the energetic phase space. And as
energy is an additive quantity one can derive an aggregated energy level function for the
entire system and test its feasibility of upscaling,

### 5.4   Concluding remarks

Overall we conclude that a thermodynamic perspective on hydrological systems provides
valuable insights which move beyond the exciting issue of using thermodynamic optimality
for uncalibrated predictions. Our most important finding is that the energy level function,
which can be seen as a straightforward generalization of the soil retention function, accounts
jointly for capillary and gravitational control on soil moisture dynamics. With this we bridge
the scale between local, non-linear soil physical controls and system level topographical
controls on storage dynamics. The beautiful outcome is that linear behavior at the system
level does not compromise the non-linearity of soil water characteristics. In contrary it may be
easily explained by the dominance of potential energy for catchments with pronounced
topography and during not too dry conditions. The option for linear behavior of hydrological
systems is inherent to Darcy's law for the unsaturated zone itself. The latter is likely the
reason why conceptual models, which usually do not account for soil physical characteristics,
work in some catchments very well and others don't. Based on the presented findings one
could speculate that conceptual models work well in system which are dominated by potential
energy, at least in the Attert experimental basin this statement is consistent with the study of
Wrede et al. (2015).

ACKNOWLEDGMENTS: This study contributes to and greatly benefited from the
"Catchments as Organized Systems" (CAOS) research unit. We sincerely thank the German
Research Foundation (DFG) for funding (FOR 1598, ZE 533/11-1, ZE 533/12-1). The authors
acknowledge support by Deutsche Forschungsgemeinschaft and the Open Access Publishing
Fund of Karlsruhe Institute of Technology (KIT). The service charges for this open access
publication have been covered by a Research Centre of the Helmholtz Association. The code
and the simulation projects underlying this study are freely available on request.





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
