# Peer review of "Energy states of soil water – a thermodynamic perspective on soil 1 water dynamics and storage controlled stream flow generation in 2 different landscapes 3 Erwin Zehe1, Ralf Loritz1, Conrad Jackisch1, Martijn Westhoff2, Axel Kleidon3, Ther"

_Hydrology and Earth System Sciences, 2018_

## Referee Comment (RC1) · Anonymous Referee #1 · 15 Aug 2018

In a nutshell: This paper has a lot of potential. However, there are important caveats that raise serious concerns. Revision is therefore strongly recommended.

A) General considerations

Overall, I sympathize very much with the good intentions of the research leading to this manuscript, and with the quest to bring more physical rigour and credibility to the hydrologic endeavours and re-discovering the primordial Earth System DNA of hydrology that is often largely absent in engineering hydrology.

From the linguistic side and overall presentation, this manuscript is well-written and moving. Were this an outreach article, I would be delighted to read this paper.

[Figure]

However, in a formal context I see red flags arising when I look at the science beneath the text. I have serious concerns on how physical principles are naïvely invoked and formulation deployed in the manuscript, and how this manuscript follows a misleading line of research promoting problematic thermodynamic considerations as if they were rigorous thermodynamic physics.

While there is some basic mathematical care in the formulations and the argumentations and schematics make heuristic sense, physically the work does not yet meet the high standards that the authors surely intend to pursue, namely in terms of physically consistency.

Even so, admittedly the manuscript provides very nice intuitive explanations about the author's interpretation of hydrologic functioning, at textbook level.

Hydrology has a long history as an Earth System science with strong principles in thermodynamics and complexity and there is nothing new in this study that advances science in that regard. This is a simple nicely written hydrology paper trying to address a highly relevant particular problem with a practical formulation and can only be duly credited as such, without any presumption of building any fundamental "theory". A theory is supposed to be more general, aiming at universality within its scope, quite unlike what is proposed.

B) Perturbation approach

The authors apply a rough reasoning from first-order perturbation theory in analytical mechanics to deploy their work and even to discuss basic nonlinearities. In doing so, their framework unwittingly requires that the Earth System is trivially forced and then released to go back to an idealized equilibrium with the freedom to follow thermodynamic optimality principles.

However, the Earth System is neither forced so simplistically nor in a position to freely relax towards equilibrium. Consequently, in practice thermodynamic optimality aspects

such as regarding entropy production or free energy decay rates are not allowed to be manifested in the hypothesised manner, which naturally reflects on the inadequacy of the simplistic functional solutions discussed in the paper.

In other words, deviations from storage equilibrium are treated in a perturbative theory setting that is only valid for small perturbations followed by unrestricted return to equilibrium. And that, sorry to say, is neither physically consistent nor useful in a real-world setting.

B) Energy currency

The authors introduce new names for trivially known concepts, like the "Energy state function" for something that is fundamentally not more than a mundane thermodynamic potential routinely used in various applications including in the Earth sciences. I wonder what justification exists for reinventing new names for already existing concepts.

Obviously, when looking at the catchment as with any other system, everything can be characterised around free energy and related thermodynamic potentials. There is nothing fundamentally novel about that and significant sectors of groundwater and broader geophysical hydrology already work in such energy currency.

C) Caveated "principles"

The thermodynamic optimality "principle" of Maximum Entropy Production (MEP) invoked in the text is not well accepted in Physics and Chemistry given that it is valid only under stringent assumptions that are not generally valid, and fundamentally discredited in those fundamental disciplines. Unfortunately though, MEP is mistakenly treated in various applied disciplines as if it were a real general thermodynamic principle.

The Maximum Entropy Production (MEP) approach is only valid as a limiting case of free flow in far from equilibrium macroscale conditions, with instability assumed to fully drive the macroscale dynamics under restrictive conditions such as local equilibrium. This may make some sense at first sight but cannot be guaranteed in complex geophysical flows.

Therefore, when the authors discuss perturbations followed by restoration towards equilibrium and invoke physical reasoning, they should purge the paper from overstating problematic principles, otherwise readers will continue to be misled and the chain of MEP disinformation will further propagate in the literature, further hampering future research in Hydrology and Earth System sciences.

D) Caveated quotes and claims

The insight from Aristotle had already been widely mentioned in various scientific fields, including Ecology and Hydrology, long before the authors and their reference did so. However, neither the authors nor their mentioned reference invoke the expression with appropriate scientific and technical rigour.

In fact, treating a system in a holistic manner as a meta-organism does not qualify for being more than the sum of the parts, i.e. system holism does not guarantee that the whole is more than the sum of the parts. The Aristotle quote is only valid in specific system categories and embarrasses the paper in the prominent way that it is presented right at the start. No extra beyond the sum of the parts is rigorously analysed or computed anywhere in this paper.

Another problem pertains the general presentation style of the manuscript with unfair claims and hype. There is an unjustified sense of self-importance in the manuscript as it addresses the state of the art about having brought Thermodynamics into Hydrology. The paper makes such allegations whilst ignoring a vast body of literature in fluvial geomorphology and ecohydrology where hydrologic problems have been treated in a sound thermodynamic manner for decades and even taught in hydrology classes for earth science degrees, and long perceived and treated as a complex system or a meta-organism as in the Gaia hypothesis and exhibiting emerging features at system level.

The classical literature came long before recent literature such as Savenije and Hrac

(2017) cited in the paper. The aforementioned opinion paper essentially reinvented the wheel with well-known generalities about Hydrology in the broader Earth System. While such unscientific hype can be digestible in tabloid-style opinion papers, it should not be admissible here where the authors worked hard to make real science. My stern advice is not to ridicule an otherwise fine work with such unscientific hype.

E) Summary and way ahead

Overall, there is merit of this manuscript in raising further awareness among engineering and statistical sectors within hydrology that hydrologic science is more than a naïve data science, and that it is fundamental to actually think about how the system is physically structured and operates like hydrology geoscientists have been doing for a long time.

I just wish that the physical arguments for the particular problem under consideration would be more consistent - hence my disappointment with this paper and my call for a thoughtful, sober revision.

As a way ahead, and this can be easily done in a sober revision:

First, please play out the cards as they are. When introducing concepts and quantities, they should be framed in a clean manner for what they really entail, rather than giving the illusion of fundamental novelty and using hypothesis that lack the validity and generality that is claimed.

When invoking "principles", it should be extensively explained to the readers what the domain of applicability, the underlying conditions, caveats and open questions are. Overall, all the assumptions and technical options, along with their limitations, need to be thoroughly justified, so that readers can approach "principles" for their real value.

Second, please eliminate the exaggeration and hype reminiscent of the literature upon which many of your arguments are based. This is supposed to be a rigorous scientific article, not a buzz-worded paper mixing technical science with rebranding old concepts

yielding the illusion of novelty, out-of-context or unjustified quotes that look good but make no sense in this paper (e.g. Aristotle), perpetuating mistakes such as misuse of ill-posed thermodynamic optimality principles and perturbation theory, and further misinformation that otherwise propagates downstream in the chain of knowledge as has been happening to the related literature. Just because mistakes are published and highly cited, that does not make them correct.

The authors have the opportunity to amend their work in a scientific-technical sense, or at least to plainly explain the limitations and caveats of their formulation so that nobody is being misled anymore.

I will trust in the authors' willingness and ability to take my concerns into consideration in producing a revised version of their manuscript. For these reasons, and notwithstanding my skepticism about the work, I see value in a modest, cool-minded version focusing on the real science which has merits.

Therefore, I would not outright dismiss this paper but rather give a second opportunity that can be achieved through reflection and revision. The authors should be given a chance to amend their work and strengthen the scientific merit of their message.

It is clear that the authors have good hydrologic insights and there is a lot to be learnt by many readers in that regard. What is critically needed now is to correct the physics, which are fundamentally flawed, or at least make the limitations straight and clear to the readers. And for that, it is crucial to look beyond.

Thank you and good luck.

---

## Author Comment (AC1) · 11 Sep 2018

On behalf of all co-authors I sincerely thank reviewer 1 for her/his thorough, eloquent and helpful assessment of our work. Please find below our reply structured along the main headlines selected by the reviewer.

General considerations A):

- We tried indeed to keep the language simple; and the analogy to an outreach paper is maybe not totally wrong. Beside our scientific objectives, we seek indeed to intro-duce a wider part of the catchment hydrological community to indeed straightforward

thermodynamic / and energy based reasoning. While we admit that a sub group of hydrologists has a background in thermodynamics – maybe not always that strong as the one the reviewer obviously has – it is our experience that a thermodynamic perspective is not so straightforward to many of our fellow colleagues from catchment hydrology.

- The proposed analysis can indeed be carried out with the knowledge of an undergraduate course in thermodynamics. The header "theory" of section 2 of the manuscript might thus indeed be misleading. But it was by no means intended to claim that the approach is fundamentally new. We will rename section 2 as "theoretical background" in the revised manuscript. However, the fact that an analysis is based on straightforward thermodynamic grounds, and might thus be understandable for a wider range of readers, does not imply that the underlying science is not helpful. In fact we provide evidence that the proposed science is helpful to inter compare storage dynamics among systems in different geological and pedological settings in an informative and illuminating way.

- It is particularly new to visualize of capillarity and gravity of soil water dynamics in a single thermodynamic potential. This defines the possible range of system states, determines whether the system is in a state of a storage excess or deficit and is helpful to visualize which part of this state space is visited real world observations. In this context our statement on the gravity control is maybe misleading. We do not doubt that soil physicists and hydrologists are aware of gravity and potential energy, but the fact that this "is a linear control" is not so frequently discussed. The major focus is often on capillarity – while this is appropriate at small systems sizes like a soil core, gravity starts to dominate in case of a pronounced topography (as shown in our study).

- In this context I wonder which a) in consistencies the reviewer of the "theory section" the reader is actually referring to and b) which the missing references are. If we missed relevant literature in this respect, we would very grateful if the reviewer provided the missing references she/he has in mind. We will happily read those and include them, in case they are relevant (I come back to this point later on).

Perturbations B)

- We absolutely do not believe that the earth system is forced in a trivial manner, with perturbations from and relaxations back to a static equilibrium. And I do not think that we claimed that the earth system is that simple in our manuscript.

- We refer to perturbations exclusively associated with soil water dynamics in the unsaturated zone. And please note that in this context we do not talk about "thermodynamic optimality". We refer to the paper of Savenije and Hrachwitz (2017) as they speculate about a storage optimum which balances recharge and release. But this is to show that this balance is a straight forward equilibrium. We will better stress this in the revised paper.

- Please note that the question whether the soil system operates linearly nor non-linearly depends on the fact whether gravity controlled potential energy dominates strongly against capillarity influences or not, because it is later which adds non-linearity. Potential energy may indeed dominate against capillary surface energy, as shown for the Colpach, and particularly if one moves upslope. I think the thermodynamic potentials we show make this very clear.

- The reviewer may forgive me, when I stick to the point that free of soil water shows indeed fluctuations around the local equilibrium – in fact we provide experimental evidence for this. Let's think about an isolated soil column in contact with a ground water reservoir. Such a column will for sure relax to local equilibrium either by means of capillary rise (if it is too dry) or by seepage loss to the GW body. After that it will remain there, because it is a maximum entropy state! The related saturation profile depends on gravity and the retention curves (this is indeed textbook knowledge). We do of course not think that this storage equilibrium is static in a real world hydrological system. It will change with tectonic changes in topography, with changes in climate regimes (affecting ground levels, as mentioned in the paper) and also which ongoing weathering of the soil. This is for sure worth to be mentioned in the revised manuscript.

However, the timescales of these processes are much larger than the timescale of the typical inter-storm period as well as of the infiltration and seepage processes in the area of interest. Also ground water levels vary on much longer times-scales. Hence, free energy of soil water does really fluctuate around the local equilibria, forced either by evaporation loss and infiltration during recharge. Note that Figure 7 and Figure 8 provide clear experimental evidence for this, at least for the Wollefsbach. (We can provide also simulation evidence if wished). In case of the Colpach the system does indeed never reach its equilibrium, but this is clearly explained in the manuscript.

- The above mentioned issue of timescales needs to be properly mentioned in the revised manuscript. And yes we will stress that the perturbations are small (particularly in the Colpach), much smaller than the straightforward physical limit defined by saturation and the residual water content.

Energy currency C)

- It is absolutely right that the proposed characteristic is a thermodynamic potential, characterizing capillary and gravity control as function of time. We will stress this in the revised manuscript. To my knowledge it is however not true that this particular potential is frequently used in hydrology – I am not aware of a single study that uses such a combination. In contrary, usually soils are characteristics by their retention curves (which are by the way also a thermodynamic potential) and topographic maps but not in the way we proposed in this study.

- With respect to the selected name – it is nice to have a name for a baby, easier to reference as the "thermodynamic potential of unsaturated zone introduced in section 2 Eq. xxx". Also the soil water retention curve is by no means more than a thermodynamic potential, not for total free energy but for capillary surface energy density. Yet the baby has a name. Alternatively one could speak about the total potential function of the soil system. We do not care too much about which name might be more appropriate, but a name is certainly helpful.

- And yes any system can be characterized around free energy and thermodynamic potentials, this is why we selected the approach, we will stress this even more. Yet I am not aware about study that uses free energy of soil water for comparing storage dynamics in catchment systems. I am keen to read any study I missed in this respect, and to include it into the reference list, if it is relevant. And yes, energy and entropy and a thermodynamic perspective is used in many fields of geo-science including hydrology. This was never denied.

Caveated principle D)

- I agree that thermodynamic optimality is by no means an accepted principle which is universally valid as for instance the second law. On the contrary thermodynamic optimality is controversially discussed and it makes rather strong assumptions for instance like steady state conditions, and close to equilibrium conditions. I agree that both assumptions are problematic in hydrological systems, particularly when talking about the intermittent perturbations of the soil system. In this passage of the introduction we sure overdid it with our will to write this paper in a smooth and simple way. We will revise this passage to put much more emphasis on these points and thank very much for this hint.

- A periodic forcing does however not imply that no MEP configuration exists. On the contrary Westhoff et al. (2014) et al showed for a simple bucket model that periodic boundary conditions lead to an MEP configuration that largely differs from the one obtained using a steady state forcing.

- Last but not least, optimality principles are quite successful in physics for instance in classical mechanics the principle of minimum action – as the reviewer does for sure know. So the quest/search for a similar principle which is valid for dissipative systems is maybe not that astonishing. And the idea that the steady state configuration of open systems coincides with a configuration that maximizes entropy production is not too awkward - also from the Gaia perspective which was mentioned by the reviewer.
Caveated principles and quotes D)

- I agree that the holistic view/approach to terrestrial ecosystems and even the earth system as a whole is much older than the opinion paper of Savenije and Hrachowitz (2017).The hint to Gaia hypothesis is very well placed in this context. We will refer to this in the revised manuscript. The reason why we selected the quote of Aristotle in the context of the study of Savenije and Hrachowitz (2017) was not to "over-mystify" but in fact to argue that the proposed balance of release and recharge is a straightforward manifestation of very basic thermodynamic laws. It was by no means our intention to embarrass anybody with this opening statement. I agree that we do not address the question whether the whole is more than the sum of it's parts in our study. So the quote is maybe indeed misplaced and we might remove it. Self-organized systems might however be more than the sum of its part, as shown by Haken with his work on laser light, his later ideas on synergy and enslavement.

- Section 1.2 "Thermodynamic reasoning in hydrology" might indeed miss a few important references on the use of thermodynamics in environmental sciences. We referred to quite a few in this context, studies in this context, after stating that "thermodynamics in catchment hydrology gained substantial attention after the work of Kleidon and Schymanski (2008)". I think this claim is correct – but I also agree that we left out quite a bit of work of other groups such as of a) the group of Amilcare Porporato (e.g. characterizing the water cycle as idealized thermodynamic cycle), b) the group V.J. Singh and the work on entropy based concepts for soil water movements and c) or most of the work of Majid Hazanizadeh, Casey Miller and Bill Gray (we referred a little bit to their work when discussing the REW concept). This has been left out not to over emphasis our own importance but simply because it was not considered as too relevant here. Be assured that I have high respect particularly for the work of the latter scientists.

To conclude, we will carefully revise the study and particularly parts of the presentation as outlined above. Let me stress again that we would very grateful if the reviewer provided the missing references she/he has in mind. We will happily read those and

include them, in case they are relevant.

Thank you very much,

Erwin Zehe

---

## Referee Comment (RC2) · G.H. de Rooij (Referee) · 19 Sep 2018

Review comments on 'Energy states of soil water - a thermodynamic perspective on storage dynamics and the underlying controls' by Erwin Zehe et al. MS No.: hess-2018-346

The paper attempts to derive a thermodynamic expression for the total potential energy of solute-free, slow-moving water in a rigid porous medium. Taking hydrostatic equilibrium as the equilibrium state of a soil it uses this expression to examine if a soil has excessive or insufficient water to see in which direction the soil water status will naturally evolve. An extensive test on two different catchments in a temperate climate

is carried out to illustrate the practical application of the equation and demonstrate how it can show the dynamics within a catchment and the differences between catchments.

Major comments

The second part of the paper is clearly the strongest, and my comments are limited to some suggestions for improvement and requests for clarifications. I was impressed by the potential of the method, and liked the way the demonstration was set up.

I am quite critical of the first part though, and the vast majority of my comments focus on that section because I believe older literature (including my own) was not acknowledged adequately, which could have saved the authors some time. I agree with the other reviewer that the presentation of that part has a certain pompousness that the contents do not justify. The second part of the paper makes clear that this is an interesting contribution to an on-going debate in the literature to which several authors of the paper have contributed over the past years, partially within the CAOS project, and to which I too have made some contributions. If the paper presents the material as the next leg of this on-going journey without too much further ado, it will be much more compelling. As a case in point, quoting Aristotle while overlooking the literature of the last century on the thermodynamics of soil water does not create a favourable impression. I provided a set of references and a detailed discussion of some of those in my detailed comments that hopefully will be of help to the authors.

Why did you choose the Helmholtz free energy instead of the Gibbs free energy? According to the formal definitions of both (Callen, 1985, p. 146-147), $J = U - TS$ and $G = U - TS + PV$, where $J$ is the Helmholtz free energy, $G$ is the Gibbs free energy, $U$ is the internal energy, $T$ is the temperature, $S$ is the entropy, $P$ is the pressure, and $V$ is the volume. Pressure seems to me to be a relevant variable, so why leave it out? The second part of the paper is quite interesting. The thermodynamical analysis leading up to it either needs to be cleaned up or removed. As I point out in several detailed comments it lacks the rigour that is required and overstates its novelty, as the other reviewer also

states. The thermodynamical analysis of soil water and groundwater was essentially completed in the 1940s to 1960s, and I provide numerous references to those works. The authors quote only Iwata (1995), whose older work highlights the variation of various thermodynamical quantities within a single pore. I recommend using the work of Bolt and colleagues for the catchment scale instead. That would allow the removal of several weaknesses in the current write-up: the scale ambiguity, the lack of terms for the interaction with the solid phase and for the groundwater pressure, the absence of a geometry term, the strange discussion of a water sphere, the reliance on cylindrical pores, and the choice for the Helmholtz free energy instead of the Gibbs free energy.

I would even go so far as to recommend to let go of the thermodynamics completely and instead adhere to the terminology adopted in all major soil physics text books in which the potential energy of solute-free soil water in a rigid soil without overburden pressure consists of gravitational potential energy and matric potential energy - kinetic energy being many orders of magnitude smaller in nearly all cases. The equations for the total potential energy and its components of a body of subsurface water of arbitrary size are already in the literature (de Rooij, 2009), and with those you can develop your entire theoretical framework without unnecessary distractions.

As stated above, I developed equations for average matric, gravitational, and hydraulic potential and average water content for large volumes, as well as their expressions for hydrostatic equilibrium and unit gradient flow, in de Rooij (2009). I also wrote in that paper that 'An upscaled h sub V ( theta sub V ) relationship according to Eqs. (19) and (20) incorporates spatial heterogeneity and allows h to vary with elevation under hydrostatic equilibrium conditions. The relationship for megascopic V (e.g., a soil layer within a field plot, or an entire field) will be of little use to calculate actual flow, but by comparing the actual h sub V and theta sub V to the equilibrium curve, the deviation from equilibrium can be asserted, and the tendency of V to absorb or release water from or to its surroundings (e.g., the groundwater, or a stream) can be established with a more or less quantitative measure.' Thus, it is clear that the central thesis of the
current paper was already formulated 9 years ago. I believe the paper as it stands does not accurately reflect this. Note that I limited the potential application to a field because assuming hydrostatic equilibrium in an entire catchment leads to a lake in the lower reaches and unrealistically dry conditions near the watershed. In this paper, this is resolved using the height over nearest drainage instead of the height with respect to a fixed reference height. This is the innovative theoretical element of the paper.

Please consult the guide for authors regarding the use of footnotes.

Detailed comments

l. 49. The authors appear to be unaware of a body of work on the thermodynamics of soil water and groundwater in the 1940s to 1960s that I believe to be highly relevant to this work. I particularly recommend the rigorous treatment by Bolt, Groenevelt, and coworkers (Bolt and Frissel, 1960; Groenevelt and Bolt, 1969). The references in these papers provide access to other papers. By not acknowledging the earlier work by several authors, the novelty of the work is overstated. It is also apparent that the level of rigour does not match that of the work around the middle of last century.

The authors quote Iwata (1995) (reference in the paper). I read Iwata's earlier work (Iwata, 1972a,b,c, 1974a,b) but found it not easy to penetrate. He argues that the thermodynamic status of the soil water depends on the distance to charged clay particles, which he treats strictly in one dimension (distance to the surface of a single clay plate) and therefore advocates to treat the soil water is a series of thin layers that are all homogeneous, but with different values for the various thermodynamic variables. Yet, when discussing the effect of the air-liquid meniscus he assumes a cylindrical pore without attempting to resolve the obvious discrepancy between the two geometries.

Iwata's line of thought culminates in the necessity to adopt the chemical potential of soil water as its true thermodynamic state, even going so far as to call the matric potential meaningless. The water pressure is a component of the chemical potential as defined by Iwata. The practical application of the concept is hampered by the necessity to

divide the water in an individual pore in an unspecified number of layers. This also made it difficult for me to fully grasp the critique on the matric potential, which can realistically only be measured at scales far beyond the pore scale. I am willing to accept a complicated interplay between various forces in the diffuse double layer that make the composition of the soil solution vary with distance from the clay surface. Nevertheless, I am also willing to accept that water flow and movement of ions over a few centimeters (Representative Elementary Volume scale – Bear and Bachmat, 1991, p.14-29) is fast enough to establish an equilibrium in which the sum of the components of the soil water potential is essentially the same everywhere, or at least exhibits a gradient that is observable at that scale. I am not sure how to reconcile that with Iwata's work, but the authors seem confident this can be done. If the thermodynamical framework remains part of the paper, an elaboration of the mathematical formalization of that reconciliation by the authors would strengthen the work, all the more since they use the matric potential themselves, albeit under another name – see my comment about l. 185.

l. 52, 57. What is the definition of power in this context?

l. 59. See also the minimization of energy dissipation during groundwater flow discussed by van der Molen (1989).

l. 84-87. The phrasing seems to suggest that the internal redistribution of water in response to external forcing (differences in rainfall, incoming and outgoing radiation, and potential evapotranspiration; gravity for sloping areas) is much faster than the fluctuations in the forcings, but is this truly the case? One could argue that the system (groundwater/soil water) is always running behind, being in a state of perpetual perturbance. How do you find the reference state in that case? Furthermore, it is quite possible that the system is still responding to a previous stimulus, and therefore seems to react to the current stimulus in the wrong way. Case in point: sunshine after heavy rain. Water should be moving upward to respond to the radiative forcing, but apart from the top few centimeters, the infiltrated rain water is still moving down. The subsurface

has a degree of inertia that increases with depth, and the inertia of an aquifer increases quadratically with its size. For simple aquifers this inertia can even be quantified by their characteristic times, which can be in the order of centuries for extensive systems (de Rooij, 2013). Without an inertia term I think the theory remains incomplete.

The matric potential is highly dissipative over distances of the order of < 1 m, but not at all for catchment-relevant distances > 102 m. One can reasonably argue that a well developed root system of an individual organism (tree) offers a dissipative mechanism over tens of meters. Less direct but not entirely baseless would be to argue that the root network of a crop or a natural ecosystem also dissipates the matric potential, even though a direct communication over the root network of an individual does not exist. In this case, the dissipation can be argued to occur from similarity of the atmospheric boundary condition experienced by the entire plant community in combination with variations in water stress, that are then evened out by their effect on the local actual evapotranspiration. In the absence of vegetation, this subsurface communication effect breaks down. The only horizontal transfer of information in that case is the reduction in the potential evapotranspiration brought about by the water transmitted to the atmosphere from moist areas. This process gives rise to the complementary relationship between actual and potential evapotranspiration (Brutsaert, 2005, p. 136-137 and references therein), and is typically considered to operate on scales much larger than that of the catchments discussed here.

l. 87. The term water stock used here is used casually, but really needs careful consideration. From Fig. 1 it is readily clear that thermodynamic equilibrium as used in this paper corresponds to hydrostatic equilibrium, i.e., a spatially uniform hydraulic head throughout the catchment. The value of this hydraulic head is a function of the water stock. Note that this function is neither unique nor monotonically increasing because of hysteresis. A catchment's water stock at any given time is a function of past external forcings, internal geomorphological processes, and the catchment biota (through canopy interception, vegetation effects on infiltration, root water uptake, creation of

macropores by flora and fauna, and a myriad of other processes). Furthermore, it varies in time. Once the water stock is determined for a catchment at a given time, and if the geohydrological make-up of a catchment is known in sufficient detail, it is possible in principle to determine the hydraulic head corresponding to hydrostatic equilibrium. Especially in sloping areas this will correspond to flooded lower reaches of the catchment and dried out streams and absent groundwater in higher elevations. You essentially have one or more lakes surrounded by a flat groundwater level and no streams. This state is therefore wholly unrealistic.

Yet you need it to determine the equilibrium groundwater level because you need to know the current status of your catchment given the amount of water that it currently holds. If the average groundwater level is lower, the average depth of the unsaturated zone is larger, and it will hold more water. By necessity that extra soil water has to replenish the groundwater, so the catchment is in the P-stage. If the average groundwater level is higher, the unsaturated zone on average is thinner and drier. The catchment is in the C-stage. I can see the logic of this, but as I explained above, catchment-wide hydrostatic equilibrium is not a useful reference state in any meaningful way. It might be of use to describe the status of soil profiles in the catchment. This ties in with my argument above that lateral exchange of water in the unsaturated zone is often small and operates at scales far smaller than that of a catchment. Later on in the paper you explain the HAND approach, which I consider to be a rather practical way to circumvent these problems. Perhaps it would be good to move that to the front, because I really got bogged down in reconciling hydrostatic equilibrium with perpetual flow in a sloping catchment. Nevertheless, the dynamic nature of the water stock remains a problem that the theory cannot easily deal with. Am I overseeing something? If so, please elaborate on this. If not, this is a limitation that should be mentioned.

There are additional problems though. I do not believe you can calculate the water fluxes entering and leaving the catchment from catchment-scale variables alone, thermodynamic or otherwise. To keep track of the catchment's water stock you therefore

have to rely on hydrological modelling, monitoring data, or both. That being the case, what is the added value of the thermodynamic description of the catchment? The water stock and the groundwater level as a function of time are crucial for determining in which state the catchment is, which in turn is key for your approach. Yet you can calculate neither of these variables with your model. The models that can also calculate the fluxes your model is supposed to calculate, so I have a hard time understanding what your model contributes. Even though I like the match with experimental data that you show later on I still find it hard to pinpoint to a meaningful addition to the hydrologist's toolbox that this research provides or at least has the potential of providing in the future. Can you tell us crisply why we need this stuff?

l. 96. How do you define (and measure) the capillary surface energy of soil water? The term suggests it is strictly limited to the potential energy arising from the gas-liquid interface.

l. 96-97. The potential energy due to matric forces is reflected in the matric potential. The gravitational potential energy is due to its elevation. I assume that the capillary surface energy is incorporated in the matric potential, which also includes the potential energy associated with the interactions along the liquid-solid interface. But if that is true, the sentence seems to have a tautology in it. Also, the qualifier 'in absolute terms' appears misplaced. Surely the sign of the difference in the two energies is crucial.

l. 98. From my comments above it is clear that I do not believe capillary surface energy differences are the dominant driver of soil water dynamics. I think the total hydraulic head is the main driver: gravitational + matric + osmotic + overburden + pneumatic potential. I do agree that under sufficiently dry conditions, the gravitational contribution becomes negligible, but I am less willing to dismiss the role of the liquid-solid interface. By ignoring it, one is unable to discuss the effect of soil wettability on the formation of preferential flow paths during infiltration, to name an extreme example.

l. 100. The energetic distance to equilibrium is not the only factor that determines the

amount of necessary recharge. The pore architecture and the phase distribution in it also factor in. Bolt and Frissel (1960) therefore included a geometry factor in their equations, which is missing in this paper. I expect that a good deal of theoretical work will go into deriving this geometry factor for an entire catchment. At the scale of a plot I analyzed a special case (rainfall while the top of the capillary fringe initially was near the soil surface) in detail in a paper (de Rooij, 2011) that was inspired by an earlier paper by members of the current group of authors and is part of the debate that the current paper continues. In the terminology of this paper, the test case was concerned with a very rapid transition from a C-regime to a P-regime. It demonstrated with how little water a large energetic distance could be overcome in mere seconds given the right circumstances. By doing so, de Rooij (2011) implicitly warned against using the energetic distance is the main criterion. The architecture of the pore space and distribution of the liquid and the gas phase in it cannot be ignored.

Fig. 1 The figure has an R-regime that I suspect should be the P-regime.

l. 118-119. Preferential flow paths by definition bypass relatively dry areas in the top soil, and provide a conduit through which infiltrating water reaches the groundwater, or at least the wetter subsoil, more quickly. Thus, a dry area of the soil is to some degree shut off the terrestrial part of the hydrological cycle. The vegetation suffers, which is why farmers mix in clay or apply surfactants to eliminate soil hydrophobicity to eliminate preferential flow. But you argue the other way: preferential flow paths accelerate recharge of the dry soil. If this were the case, would farmers not promote hydrophobicity instead, to encourage the development of preferential flow paths? p. 6, footnote. Not only do you not consider volumetric change of the pore space, you assume a rigid soil. No deformation of any kind is permitted.

l. 151 (Eq. 1) From this equation it becomes clear that the term 'potential energy' only referred to the gravitational potential energy of the water. This goes against established terminology in the soil physics literature. Equation (1) lacks terms for the external pressure, for the forces resulting from the contact with the solid phase, and for the ionic

composition of the soil solution. Furthermore, the authors do not state if the equation is local (microscopic) or applies to the entire water phase. The text is ambiguous in that respect. Bolt and Frissel (1960) present the full equations for both cases, but for the Gibbs free energy instead of the Helmholtz free energy.

l. 164-183. This derivation is only valid if the water is stored in radially symmetric pores that are fully saturated with water behind the air-liquid interface, because the two principal radii of curvature are set to be equal to one another. Pendular rings and water films on pore walls or in corners are not covered by this derivation.

l. 177 (Eq. 5). A further simplification appears in Eq. (5), where, for reasons that are not explained, the water is assumed to occur in a sphere, which would case the water pressure to be higher than the atmospheric pressure, which is not typically the case in an unsaturated hydrophyllic soil. A more logical approach would have been to calculate the volume in the cup-shaped air pocket enclosed by the liquid-air meniscus and the plane through the air-solid-liquid contact line at its outer boundary, and see how that volume changes when the pressure difference across the interface changes. This would then have to be done for different pore sizes to find something meaningful at the sample scale and all scales beyond that. Defining the populations of pore sizes and the required range of pressure variations for which to carry out the calculations may be conceivable at the sample scale but on first sight seems to be daunting for the catchment scale.

This simplification is too extreme for the result to be meaningful I believe. The geometry factor introduced 58 years ago by Bolt and Frissel (1960) seems to offer a better starting point.

l. 182 (Eq. 6) The end result of the derivation in Eq. (6b) is the conventional expression for the hydraulic head (multiplied by rho g to obtain the potential with units Nmˆ-2 (Jury and Horton, 2004, p. 54)) if only the gravitational force and the matric forces in the vadose zone are accounted for and kinetic energy can be ignored. You can arrive there by

a Newtonian analysis of the forces acting on the water in a much more straightforward way than presented here. I believe your initial reliance on the interface being a section of a sphere made you arrive at a correct result despite requiring water to be residing in a sphere. You made two assumptions that happened to cancel each other out. But because the pressure difference between the liquid and the air is opposite because the direction of curvature of a spherical water droplet and water in a axisymmetric pore is opposite, I suspect you dropped or added a minus sign along the way.

l. 185. The term matric potential is well-defined and has been in use for decades. Why do you want to change it to the much less accurate capillary surface energy?

l. 186-188. In the analysis above you included the effect of a single meniscus. Implicitly, you defined the control volume to be so small as to envelope only a single pore. Here you invoke the continuity approach in which the macroscopic water content can be defined, which requires the control volume to be the size of the representative elementary volume, i.e. a very large number of pores (Bear and Bachmat, 1991, p. 14-29). A rigorous treatment can be carried out for the scale at which the phases can be separated or at the continuum scale, but not by switching from the one to the other. Bolt and Frissel (1960) developed equations for both analyses.

l. 192 (Eq. 7). From thermodynamics, the expansion of dV sub theta is familiar, but what does the term theta dV signify here? Your explanation only holds over incremental changes because the water content is held constant, so your phrase 'moving up scale' appears to be too broad. In thermodynamics the differentials typically stem from external inputs/outputs of energy or work done by the system or being done to the system. I do not see how any of these possibilities lead to a change of volume in a rigid soil.

l. 194. It is called the Richards equation.

l. 207-217. These insights are not particularly new. Also, the definition of linear behaviour is ambiguous as the driver of the behaviour and the response are not clearly defined. I presume one could adopt the uptake/release of water with a change in the

total hydraulic potential for different combinations of gravitational and matric potential energy. In that case, the capillary fringe creates a very non-linear trajectory even when gravity is supposed to dominate: zero response when the matric potential is larger (less negative) than the air-entry value, and a significant response when it is just below the air-entry value. In hydrophobic soils things become even weirder.

l. 258 deriving -> taking the derivative

Section 2.2 Section 2.2 does not offer any new insights in my view, but it phrases it in an unconventional way. With the rather large depth to groundwater deployed in the discussion, the limitations of the minimum free energy concepts at these scales become clear. Have the authors ever measured hydrostatic equilibrium in such deep profiles? The only area where the assumption of hydrostatic equilibrium is applied is in the sub-sealevel part of the Netherlands, where the groundwater is maintained at about 1.5 m depth. At the end of winter/early spring, when rain is not so frequent, the agricultural soils are still barren, and the potential evapotranspiration is very low, the soil may approximate hydrostatic equilibrium reasonably well. When the groundwater is deeper, a good portion of the soil profile below the root zone may exhibit near-constant unit gradient flow, during which the matric potential is constant with depth, and the purely gravity driven flow occurs at a water content for which the hydraulic conductivity equals the long-time average net drainage rate. This does not meet your minimum free energy criterion but nevertheless seems to reflect a relaxed state in which the soil hydrology has adapted to external forcings.

Section 2.3. You present families of curves at equidistant intervals of elevation above nearest drainage (HAND). You can take this one step further by characterizing a catchment through the probability density function of HAND, dividing the range in intervals with equal probability mass and then plot the curves that are representative for each of these intervals. These plots will then convey useful information about the catchment that the current Fig. 4 does not.

l. 349. In most climates, the total fraction of time with rainfall is much smaller than the complementary fraction without rain. The only exceptions that I can think of are cloud forests. Furthermore, rainfall involves generally higher flux rates as evapotranspiration, and are not influenced by the soil moisture status of the soil at the time the rain is falling. Finally, wet soils are more conductive than dry soils and therefore can move back to hydrostatic equilibrium more quickly than dry soils. Without formal analysis we therefore can conclude that the C-regime occurs more often than the P-regime in the top soil. The deeper we get, these fluctuations between wetting and drying are damped ever more strongly. At some depth that depends on soil properties and climatic conditions, a nearly constant downward flow occurs, with a vertically uniform matric potential. Within 1-2 m from the groundwater level the matric potential profile responds to the phreatic level. The section of soil under unit gradient conditions is permanently in the P-regime, I believe. Depending on the flux density this is somewhere between hydrostatic equilibrium and unit-gradient flow. This seems to imply that the installation depth of your tensiometers/water content sensors may affect the status that you assign to the free-energy regime of the catchment. I would like to see that discussed in the paper.

l. 375. Optioned?

l. 375-377. I am not convinced that fitting a single retention curve through the data points of a number of samples gives you the catchment-scale retention curve. Of prime concern is the conservation of mass when moving from the sample scale to the catchment scale (de Rooij, 2009, 2010). This dictates that the hypothetical representative soil profile should have the average depth (distance between the soil surface and the phreatic level), and that its saturated water content equals the weighted average of that of the samples, with the weighting factors equal to the fraction of the catchment soil volume represented by the individual sample. If one prefers to have the catchment-scale groundwater table at a different depth for whatever reason, the saturated water content must be multiplied by the ratio between the average groundwater depth and

this chosen depth to ensure mass conservation. Furthermore, if all soils are saturated, the catchment as a whole will start losing water when the first sample reaches the air-entry value and upon further drying will stop losing water when the final sample is dry. (The fact that the van Genuchten function captures neither of these points will be saved for another day.) The latter requirement is of no practical value because this situation will never be reached. Complete saturation is also unlikely, but the catchment can realistically come close, for instance after prolonged rain shortly after snow melt. Thus, mass conservation forces us to fix the catchment-scale saturated water content and its air-entry value at a weighted average of all samples and the value of a single sample, respectively. Optimizing these parameter values based on goodness of fit will lead to mass balance discrepancies. If one devotes some thought to the matter, other critical matric potential levels can probably be defined. The catchment scale water content at hydrostatic equilibrium should be correctly represented by the catchment-scale retention curve as well. Perhaps unit gradient conditions at different matric potentials can also provide useful values. Straight-forward parameter fitting is not the best approach for 'what the authors have in mind.

Note that in Fig. 5 (l. 393-394) you state that you used energy-conserving averaging, which seems to contradict the text. Can you give the equations of the averaging operation?

l. 430 Is an observations the sum of an observed matric potential and an elevation multiplied by the locally observed water content?

REFERENCES

Bear, J., and Bachmat, Y.: Introduction to modelling of transport phenomena in porous media. Kluwer Academic Publishers, Dordrecht, The Netherlands, 1991.

Bolt, G. H., and Frissel M. J.: Thermodynamics of soil moisture, Netherlands journal of agricultural science, 8, 57-78, 1960.

Brutsaert, W.: Hydrology. An introduction, Cambridge University Press, Cambridge, UK, 2005.

Callen, H. B.: Thermodynamics and an introduction to thermostatics, 2nd ed., John Wiley and Sons, NewYork, U.S.A., 1985.

de Rooij, G. H.: Averaging hydraulic head, pressure head, and gravitational head in subsurface hydrology, and implications for averaged fluxes, and hydraulic conductivity, Hydrol. Earth Syst. Sci., 13, 1123-1132, 2009.

de Rooij, G. H.: Comments on 'Improving the numerical simulation of soil moisture-based Richards equation for land models with a deep or shallow water table', J. Hydrometeorology, 11, 1044-1050, doi: 10.1175/2010JHM1189.1, 2010.

de Rooij, G. H.: Averaged water potentials in soil water and groundwater, and their connection to menisci in soil pores, field-scale flow phenomena, and simple groundwater flows, Hydrol. Earth Syst. Sci., 15, 1601–1614, doi: 10.5194/hess-15-1601-2011, 2011.

de Rooij, G. H.: Aquifer-scale flow equations as generalized linear reservoir models for strip and circular aquifers: Links between the Darcian and the aquifer scale, Water Resour. Res., 49, 8605-8615, doi:10.1002/2013WR014873, 2013.

Groenevelt, P. H., and Bolt, G. H.: Non-equilibrium thermodynamics of the soil-water system, J. Hydrol. 7, 358-388, 1969.

Iwata, S.: Thermodynamics of soil water: I. The energy concept of soil water, Soil Science 113, 162-166, 1972.

Iwata, S.: Thermodynamics of soil water: 2. The internal energy and entropy of soil water, Soil Science 113, 313-316, 1972.

Iwata, S.: On the definition of soil water potentials as proposed by the I.S.S.S. in 1963, Soil Science 114, 88-92, 1972.

Iwata, S.: Thermodynamics of soil water: III. The distribution of cations in a solution in contact with a charged surface of clay, Soil Science 117, 87-93, 1974.

Iwata, S.: Thermodynamics of soil water: IV. chemical potential of soil water, Soil Science 117, 135-139, 1974.

Jury, W. A., and Horton, R.: Soil physics. 6th ed. John Wiley and Sons, Hoboken, NJ, U.S.A., 2004.

van der Molen, W. H.: Physics versus mathematics in groundwater flow – a physical explanation of the minimum theorem in finite element calculations. J. Hydrol. 109, 387-388, 1989.

---

## Author Comment (AC2) · 1 Oct 2018

I sincerely thank Gerrit de Rooij for his thoughtful and detailed assessment of our work. I am sure this consumed a considerable amount of time and this investment is highly appreciated. Gerrit de Rooij's critique is mainly targeted towards sections 1 and 2 (and a little about section 3). Before I address those points in detail I would like to clarify a few issues dealing with the usefulness of the proposed approach, its predictive potential, present modelling evidence (including spatial averaging).

The added value of our approach

Figure 1 shows the time series of soil moisture observations observed in the Colpach (blue lines) and the Wollefsbach (red lines). It is interesting to see that the ranks of quite a few sensors are stable in time. Why? Because their spatial variability is not purely random! Or course the soils in both areas are not homogeneous, hence these differences cannot be attributed to HAND differences alone (i.e. perturbations around different equilibria). I guess we see a mixture of HAND and variability in texture.

[Figure]

Figure 1: Soil moisture time series in the Colpach (left panel) and the Wollefsbach (right panel).

Despite of this interesting finding a comparison between both samples appears a little messy, and one would again conclude that the Wollefsbach is wetter than the Colpach. The comparison we presented Figure 8 of our manuscript, corroborates that quite a bit of this variability among the different sites is indeed systematic and explainable from the energetic point of few. And again we see that the Colpach operates in the P-Regime and while the Wollefsbach drops into the C-Regime.

Does it help to explain runoff generation in the catchment?

This was asked by Thilo Streck when I discussed the approach recently on a workshop of the German Soil Science Society. While Thilo found the approach "interesting" he pointed out that runoff response to rainfall is not generated everywhere the catchment, which means that the energy state at remote locations might be pretty unimportant for this. This is of course very true and today we know that during most of the rainfall events, runoff is mainly produced in the riparian zone (unless it becomes extreme and the hillslope switch on). Figure 2 shows the free energy state of an observation site which is located in the riparian zone of the Colpach (there are unfortunately only 2 sites).

[Figure]

Figure 2: Discharge at gauge Colpach (19.km$^2$) plotted against the energy state of soil moisture in the riparian zone

Note the threshold character: discharge for efree < 0 (storage deficit) is small while it shows a enormous spread when efree > 0. The spreading reflects the differences in rainfall forcing. I think this plot corroborates that the energy state of the soil water content in the riperian zone is a nice predictor to characterise the onset of rainfall runoff production in the riperian zone and it corroborates that the distance to the equilibrium provides valuable information, and provides a theoretical background to predict the onset of enhanced runoff production.

Modelling evidence

Figure 3 compares the energy states of soil moisture observed that the two selected sites discussed in the manuscript with those from simulations with a 2 dimensional physically based model (based on the Richards equation and more).

[Figure]

Figure 3: Free energy of observed (Panel a) and simulated (Panel b) soil moisture time series at the selected sites in the Colpach and the Wollefsbach catchment.

The model has been parameterized for both catchments using extensive data and it was shown to predict the water balance and runoff dynamics of at least the Colpach very well as explained in Loritz et al. (2017). Please note that for the Colpach simulations and observations are in a good accordance (the mismatch is of gradual nature), while simulations cannot reproduce the drop into the C-Regime during the summer period in the Wollefsbach. This is because the model systematically overestimates soil moisture in these soils during dry spells. I think this is a nice example that the approach is helpful to discriminate substantial from gradual model errors.

[Figure]

Figure 4: Observed and simulated free energy of the soil water stock in their corresponding energy level function (Panel a and b). Panel c shows the corresponding dynamics of the averaged simulated and observed free energies as function of saturation.

Figure 4 panel a) and b) compare observed against the simulated free energy states of the soil water stocks in energy level functions of the Colpach. Observations spread across a much wider range than the simulations. This mismatch is easily explained by the facts that the model setup of Loritz et al. (2017) does neither account for small scale soil heterogeneity, nor for larger scale variability of rainfall or of forest vegetation. We tested whether the model setup is at least capable to predict the average free energy state of the observed soil water stock. To this end we averaged the free energy states of the observed soil moisture time series and plotted those as a function of average soil saturation into the corresponding average energy state curve (Figure 4c, black circles) and compared those deviations to the average simulated free energy states (Figure 4c brown circles). Though the matching is much better for the averaged dynamics, the simulation shows a slight positive bias of 0.6 m ($5.9*10^3$ Jm$^{-3}$).

The presented model evidence underpins a) the general idea discussed in the manuscript well and b) that, due to the additive nature of energy, an inter-comparison of averaged energy states in the averaged energy state function is straightforward and helpful.

In summary think that the above presented figures and arguments are helpful to underpin the value of proposed perspective.

In the following I'll address the points brought up by Gerrit de Rooij. Before I go into the details I'd like to stress that we indeed missed to refer to relevant literature, including Gerrit's work. This was not done on purpose and we will happily correct for this in the revised manuscript. I will also not repeat our reply to reviewer 1 in respect to the quote of Aristotle and the language issue here. I guess this goes pretty much into the direction of what Gerrit de Rooij would expect. Last not least I' d like to share that I learnt my thermodynamics from a different sample of textbooks, which are usually used in theoretical physics (Honerkamp and Roemer) – I am usually a little hesitant to refer to those in hydrological papers, but this explains maybe our different background with respect to thermodynamic textbooks. Now let me come to the details.

Why Helmholtz instead of Gibbs free energy (and our understanding of potential energy).

This is because I thought that it doesn't make too much of a difference. Thermodynamics provides a large set of thermodynamic potentials: inner energy, free energy, enthalpy and free enthalpy (the terms I learned in German). At the end of the day they all have the same current (Joule) and their respective use is strongly related to thermodynamic processes in connection with "reservoirs" and to the set of state variable which are controlled during these processes.

U is inner energy and $dU = TdS$ (heat) $- PdV$ (mechanical work) $+ \mu dN$ (chemical energy, where N is number of molecules in the gas and $\mu$ chemical potential, alternatively we can use the mass M, express $\mu$ not as energy per molecule but per mass. Note this is not a total differential, as changes in T, p and $\mu$ are not considered). Gerrit is correct that the way from U to one of the other potentials is by the Legendre transformations he showed. This relates to the degrees of freedom and what is controlled in the system. Helmholtz free energy is $E = U - TS$, and assumes that the system is not in contact with a heat reservoir, hence temperature is not controlled.

$dE = -SdT$ (also heat) $- PdV$ (mechanical work as defined by Boltzmann) $+ \mu dN$ (chemical energy, or molar free energy after Bolt and Frissel). We preferred this because we treat the soil as rigid. Hence $dV_{soil} = 0$ which means we can neglect mechanical work. We agree that if we select G we end up with $dG = -SdT$ (heat) $- VdP$ (I learned this is also work) $+ \mu dN$, the latter might be interesting in the aquifer. So maybe this is the next step when looking at both the aquifer and the unsaturated zone. I admit that Bolt and Frissel account for an additional work term (which relates to the scalar produce of forces and displacements).

So if we focus on the change of free energy associated with changes in soil water content (and neglect their effect on soil temperature and use the mass related formulation we end with $dE = \mu dM$. One way to relate this term to the matric potential, is to relate it to chemical energy (as proposed by Kleidon and Schimansky (2008) or Hildebrandt et al. (2017)). I come back to this below.

Potential energy does not belong to inner energy and I also think that it is misleading to relate the matric potential (which I like very much) to potential energy. To my understanding potential energy

relates to the position of an object/body in the gravity field (which does not involve any contact of bodies or fluids) or alternatively an electrical charge in an electrostatic field, or a Hadron in the potential well of the nucleus. I think relation to an elementary force field is essential here. And note potential energy increases at a given distance if when increasing the mass of the object/body, but is does not change if we deform the body (and change the pore) while preserving the total mass.

The energy form which relates to the matric potential has totally different invariances. The matric potential energy (I never came across this term) needs contact of a fluid with a solid surface. It would not vanish if gravity were switched off. It would however change when we mix the fluid with soap (at a constant mass), it would change if we deform the soil and change the pore size distribution (without changing the stored water mass) and it changes with temperature. Last but not least it does not increase but it decreases when we add mass to the system. This is why I think that it is not correct to treat it as potential energy - it clearly depends on the inner state and inner architecture of the system, so it belongs to inner energy! I regret when this statement is against the usual terminology of soil physics, but I think it makes much sense.

One option is to relate the matric potential to chemical energy, as done in a couple of studies cited above. This implies that the product of matric potential and gravitational acceleration g is then regarded as a chemical potential. We then end up with:

$dE = \mu dM + gz dM = (\rho g \psi d\theta + \rho g z d\theta)*V \rightarrow de = \rho g \psi d\theta + \rho g z d\theta$ (with e=E/V).

The point which is unclear is how to define the total chemical energy within a finite volume of the soil. At least Axel Kleidon and me were undecided about that. His idea was to integrate $de_{cap} = \rho g \psi d\theta$ from the residual water to the actual soil water content! This is consistent with the idea of a chemical potential, however, this implies that matric potential energy will increase with increasing soil water content and peak at saturation (note this is a definite integral). This is however, not what we measure and observe with the matric potential.

The other option is to postulate that we are in a similar comfortable situation as for potential energy. Here we can not only characterize it's change in but the total content of potential energy by integrating $\rho g z \theta$ the over the volume of interest (as for instance shown by Gerrit in one of his papers). This implies that

$\partial e_{free}/\partial V = \rho g \psi \theta + \rho g z \theta$.

and that we have to integrate the right hand side over the volume of interest to obtain the total energy of the water stored in this volume. I agree with Gerrit that we could easily start with this expression and proceed.

Yet is remains unclear what the baby is and how to name it. This is the reason for our effort to relate it to surface energy (which is in fact close to what Bolt and Frissel did). The rest is known from the manuscript. I agree that the assumption of cylindrical pores is a strong simplification, though it provides a link between matric potential and surface tension and thus surface energy. The reasons why we referred to Iwata is mainly because of the need to relate the change in surface area to a change in stored and thus of soil water content.

After a look to the avenue proposed by Bott and Frissel I agree that a start from Gibbs free energy has quite an advantage. They included the surface energy term into the VdP in their micro approach

(their Eq. 6.5). This avoids the assumption of a cylindrical pore. (Yet also they treat what we call potential energy as separate term.) But their expression is dEpot= MGdh, which would corresponds in our terminology dEpot= $\rho g\theta Vdz$), is much better suited for our argumentation and simplifies the entire derivation.

We will definitely revise the theoretical background section in line with Bolt and Frissel. I also agree that a more detailed discussion of the approach in comparison with your work is very much appropriate. But please note that we do not average. In fact we stratify observations along HAND difference and we account for variability among those, we can account for variability of retention curves as well (if known). The issue of the paper is also not about effective models, but to explain differences between observations. We will better stress this from the beginning and the above figure 2, shows the relevance for discharge and Figure 3 and 4 show that distributed modeling provides similar findings.

Language issues and overstating the novelty of the work

I now realize much better that the opening statement with a reference to the recent opinion paper of Savenije and Hrachowitz (2017) and the statement that "substantial attention in catchment hydrology since the work of Kleidon and Schymanski (2008)" might appear a little strange, as the main authors of these papers co-author the present manuscript. In this respect I'd like to stress this is my fault. I wrote most of the introduction before I started the discussion with Hubert Savenije and Axel Kleidon about the approach presented. So while the tenor of these passages does certainly reflect my personal point of view, a revision of the passages is certainly appropriate.

As already stated in our response to reviewer 1 it was not intended to overstate the novelty of the work. In fact in a former version of the manuscript it added several times phrase like "as it is well known". A few of my co-authors argued thermodynamics is not well known to most of the catchment hydrologists. Nevertheless we will revise this part and add the missing discussion of Gerrit de Roojs work.

Power = Q * $\Delta$P (Q flow, P pressure)

The issue of relaxation times and time scales

Of course the soil runs behind these disturbances, yet they propagate into larger depths (and this is dissipative as well). With respect to relaxation times scales Figure 5 compares the soil water content in spring with the corresponding free energy state of a site in the Wollefsbach, note that $e_{free}$ gets negative when the soil water content drops below 0.364 $m^3m^{-3}$ (dashed line).

The time to drop into the negative range of $e_{free}$ is 3-5 days, while the switching back to the positive range at day 65 occurs within a few hours during a rainfall event. It is not too much of a surprise that the time scale for drying the soil is much slower, as it is controlled by ET and especially root water uptake and hydraulic conductivity decreases with soil water content. Wetting is much faster, also when the soil is dry. This is due to the presence of preferential flow paths, which accelerate recharge and relaxation from dry conditions and export of excess water. (We have considerable experimental evidence about those in the catchments of interest and their dominance during recharge and runoff events (Angermann et al. 2018, Jackisch et al. 2018)). Figure 5 corroborates that relaxation times are fast enough to allow for perturbations around the local equilibrium state. We will better explain that drying and optional switching to the C-Regime operates at much longer time scales than wetting (and

it needs roots to establish this perturbation in soil) and that the optional switch into the C-regime occurs at the weekly and not the event time scale.

[Figure]

Figure 5: Free energy state and soil water content during a period of a month.

We have additional experimental evidence (joint observations of the soil water content and the matric potential) which corroborates that the energy state of soil water shows indeed perturbations around the energy minimum. This has been presented at EGU 2018 (Jackisch et al. 2018), which is currently on the way to a manuscript. And we have evidence that the approach allows indeed an reasonable estimation of depth to GW, based on known the retention function and the pair of matric potential and soil water content (as pointed out in the discussion of the manuscript). I see the point with the inertia, and I am very sure that aquifers operate and respond at much larger time scales. But I think there is no mechanical inertia in unsaturated soil water dynamics. The latter would imply kinetic energy or in other words an advection term in the Richards equation. But kinetic energy is as good as zero. I am also not aware of pressure that might propagate through the vadose zone (which requires inertia as well).

Casual use of water stock and thermodynamic equilibrium

I agree with Gerrit that we used the term water stock in a casual way, in fact we should use the soil moisture stock because this is the focus. I did not see that we doubt that the soil water stock depends on the all these influences highlighted by Gerrit (we will stress the role of vegetation as stated above). Yet if we want to characterize those we are thrown back to our observables: these are time series of matric potential, soil water content and if available depth to GW, and local retention properties. (We have no salinity measurements here (to infer on osmotic potentials) and I have no idea how to assess root water potential at all). If we combine these observables as suggested in our study, we learn much more about the temporal dynamics of the soil system at different locations in the catchment. This is not the entire picture but it is a way ahead (which I never saw elsewhere in this form).

I would say it the other way around, hydraulic equilibrium is a thermodynamic equilibrium, which corresponds to a state of minimum free energy/maximum entropy! The latter implies a zero

potential gradient (including root water potential as well, we will state that we do not address this) in all directions.

In our study we do indeed focus on vertical distances, or more precisely HAND. I agree that this should be better stressed in the beginning. And yes we will be more precise in the sense that we look at the equilibrium of a distributed set of soil profiles spread along a HAND gradient. This is not the stock within the entire system, but it is a way to pool data, which are otherwise treated in a largely independent manner, into a joint sample (and explain differences) and to step beyond a mere look on comparing stored volumes!

Maybe we need to reformulate the story, as the proposed approach sheds light on whether the soil reaches the equilibrium or not. This is the case on the seasonal scale in the Wollefsbach (and the Weiherbach) but not in the Colpach.

Calculations of water fluxes from averaged quantities

I absolutely agree that you cannot calculate useful fluxes from averaged quantities without models and we will clarify this in the manuscript even stronger. I do not think that we ever claimed this, and in fact I think this is a key challenge to the REW approach. But as shown in the Figures presented above our approach can be used to inter-compare whether model errors are of gradual nature or whether the error is substantial (compare Figure 3), and to do this in a volume averaged manner as well.

[Figure]

Figure 6: Sensitivity to changes in macropore conductance

And I think way to visualize soil water dynamics through its free energy state in the energy state functions /total potential functions is helpful to explore the sensitivity to changes in system characteristics. This is corroborated by Figure 6 were we compare the simulations from the reference

model setups described in Loritz et al. (2017) with those where the conductance of the macropores in the system has been increased from $10^{-3}$ m/s to $10^{-2}$ m/s.

In both model setups the variability is reduced. The Colpach operates on average closer to the equilibrium states (due to an enhanced drainage efficiency) while the Wollefsbach operates at on average higher energy states, due to the enlarged recharge capacity.

Definition of the capillary surface energy?

See above, yes this fully relates to matric potential, but I stick to the point that this is not potential energy.

The energetic distance to equilibrium

This is an interesting point and in fact I started already to dig into Bolt and Frissel. And yes there is not much water needed to trigger strong changes in the energy state of the soil, this depends on the slope of the retention curve. Yet I think that our approach is valid, because it helps to balance this effect against the straight forward linear increase of potential energy (where the slope increases with HAND). There are breakeven points where either the one or the other dominates either when increasing saturation (this is well known) or when increasing HAND at constant saturation. It is novel to combine these two in a single characteristic. And this breakeven point corresponds to the threshold to trigger runoff response (if we focus on the riparian zone, compare Figure 2).

And yes the architecture of the pore space cannot be ingnored – so in case we go for volumetric averaging I would (for whatever purpose) I would rely on a calibrated model, which can account for this. The approach we propose can in fact account for variability in retention curves, if those were measured at the locations of the soil water sensors – but this is only partly the case.

Figure 1

Correct, I guess we remove it anyway.

Preferential flow and wetting (L 118 -119)

Yes I think that preferential flow for instance through a worm burrow, a plant root, or through does facilitate the recharge of dry soils. And there is plenty of evidence for that, for instance through reduction of overland flow production in Hortonian landscape if cracks emerge (Zehe et al. 2007) or macropore density increases (Zehe et al. 2005) or by deep routing vegetation in savannahs (Tietjen et al. 2009). If the flow path ends within the unsaturated zone, there is infiltration from the macropore into the surrounding soil. This contributes to wetting from the side (Beven and Clarke, 1986; Weiler and Naef, 2003; Klaus et al., 2013).

The derivation of our expression of free energy of soil water (lines 151 – 187)

Yes I think $e_{cap}$ is not a potential energy- note that Bolt and Frissel in their eq. 4.1 account for potential energy as separate term as well.

In a previous version of the manuscript we started with equation 6. While I thought the link between matric potential and energy is straightforward, my co-authors disagreed and recommended a theory section to help those readers without a thermodynamic background through the argumentation. I

still think that such a section is needed, but I agree the avenue of Bolt and Frissel is superior, because it avoids the assumption of cylindrical pores and includes the wetting angle.

I do not think that we recommended to forget about the matric potential. It is the other way around we recommend collection of joint data sets (matric potential and soil water content) as the product of water content and total potential provides more information about the state of the soil (and it is additive and can thus be averaged to larger spatial extents). Last not least the energetic view is also interesting opportunities to fit retention curves, as will be shown by Conrad Jackisch in a forth coming manuscript.

And I do not like the idea of a Newtonian analysis – simply as this comes from classical mechanics and thus essentially conservative systems. Friction and dissipation comes somehow through the back door. This may create a lot of misunderstanding (you may consult the argument between Ciaran Harman and myself in the discussion section of Zehe et al (2013)). Catchments and soils are open dissipative systems – the soil is so highly dissipative that soil water flows have almost no kinetic energy. Hence thermodynamics provides the right framework as dissipation is at its core.

Non-linearity and upscaling

Yes let's be precise with the issue of linearity: It is of course not new, but in my community often forgotten. The Darcy law has two sources of nonlinearity, one in the gradient term and the other in the hydraulic conductivity term (though I prefer flow resistance). In our study we just refer to the second and show that there is an breakeven point at which the gradient is dominated by gravity, which implies that free energy scales linearly with local soil water content. We could of course get this out by looking at the total potential as well. But not so nice instructive and potentials are not additive. Note that this dominance of the linear part emerges at sufficient high values of HAND, we will stress this. I am sure that Gerrit is true with his thoughts on the capillary fringe, but in fact we move to higher Hand distances here. So I do not think that this contradicts our findings.

And let's be precise with scales as well. By moving up scale we meant one hand that we increase the spatial extent of the domain by including more observation sites.  Note that this does not include volume averaging, but that we stratify the observations along lines of similar geo-potential and saturation and plot them into the respective thermodynamic potential we call energy state function.

Of course we also included the point of volume averaging (which works see above) and this has in fact already been done in Zehe et al. (2013). Yet I prefer this stratified view against the average.

Section 2.3 and energy state curves are non-informative

I disagree: these curves provide much information about different drivers on soil water dynamics and how they change with HAND and with saturation. And they are very useful for inter-comparing soil moisture observations in a meaningful stratified manner, and distinction between the ranges of efree > 0 and efree < 0 is helpful to infer on the threshold for runoff generation (Figure 2). I never saw a graph that connects topography and retention is a meaningful and stratified manner. Many things I saw rely on averaging and I doubt whether this has any value.

I agree with Gerrit that a corresponding plot for representative intervals of HAND might be better and we try this out (maybe with different line sizes to represent their fraction).

Do this perturbations across the equilibrium really occur?

They do, (see above) but not in all systems (not in the Colpach, not in Malallcahuello) and not on the event but on the weekly and seasonal scale. We will stress in the revised manuscript.

The representative retention function

I stick to this point and we additional have evidence for that. It is well known that retention in a fist size soil sample is not representative for heterogeneous systems (as the REV is much larger). So to me a set of experiments performed at 63 distributed soil cores yields a statistical distribution: As we control tension in these experiments, we observe frequency distribution of h ($\theta$) conditioned by the tension. It makes absolutely not much sense to fit relations to single experiments and average the parameter afterwards (as it is done e.g. when estimating pedo-transferfunctions). The average parameter set will not represent the average relation between soil water content and tension. It makes sense to create a representative data set, and here we do this by averaging across all water contents of the experiments at a given tension (as given below).

[Figure]

Figure 7a: Effective retention function of the Colpach

[Figure]

Figure 7b: Retention functions of obtained for single experiments

This corresponding retention function performs well in a simulation of the water balance of the Colpach using CATFLOW (find below the plot of total specific runoff versus specific discharge at the catchment outlet).

[Figure]

Figure 8: Simulated total specific runoff versus specific discharge at the catchment outlet (lower panel) and precipitation (upper panel) in the hydrological year 2013/14.

Please note that simulated runoff is strongly controlled by the wetness of local pools in the bedrock (both in the model and in the Colpach). Hence, simulated runoff is highly sensitive to the choice of the retention function as shown by the Figure 9.

[Figure]

Figure 9: Sensitivity of double mass curve to the choice of the retention function (accumulated runoff plotted against accumulated precipitation in the hydrological year 13/14). The light panel shows the NSE for the randomly selected point curves (blue), the averaged van Genuchten parameters (green), the best model setup including the effective retention curve, the same configuration without vertical and lateral macropores. Note that this is unpublished work)

Yes observations is (matric potential + HAND) * theta.

Thank you very much for the stimulating discussion and the helpful comments

Erwin Zehe

References not given in the manuscript:

BEVEN, K. J., and CLARKE, R. T.: On the variation of infiltration into a homogeneous soil matrix containing a population of macropores, Water Resources Research, 22, 383-388, 1986.

Honerkamp, Josef und Römer, Hartmann. Klassische Theoretische Physik, Springer.

Conrad Jackisch, Axel Kleidon, Ralf Loritz, and Erwin Zehe; A thermodynamic interpretation of soil water retention and dynamics. Oral presentation at EGU General Assembly 2018. Geophysical Research Abstracts Vol. 20, EGU2018-12442, 2018

Klaus, J., Zehe, E., Elsner, M., Kulls, C., and McDonnell, J. J.: Macropore flow of old water revisited: Experimental insights from a tile-drained hillslope, Hydrology And Earth System Sciences, 17, 103-118, 10.5194/hess-17-103-2013, 2013.

Weiler, M., and Naef, F.: Simulating surface and subsurface initiation of macropore flow, Journal of Hydrology, 273, 139-154, 2003.

---

## Author Response (AR1)

Dear Editor, let me first of all thank you for providing us extra time to complete the revisions. Let me furthermore thank both reviewers for their valuable points. Particularly Gerrit de Rooijs review was extremely helpful. In line with our responses to the reviewers we considerably revised our manuscript as outlined below:

- Title: We changed the title to "Energy states of soil water – a thermodynamic perspective on soil water dynamics and storage controlled stream flow generation in different landscapes". to better reflect the manuscript content.
- Section 1 (Introduction) was completely revised:
  - Following the recommendations of both reviewers we referred to relevant studies that previously used thermodynamic reasoning in hydrology and related earth sciences.
  - As recommended by reviewer 1 we now clearly state that thermodynamic optimality is controversial and that it should not be mistaken with a first principle. But we stick to the point that it might be a useful constraint to explain system behaviour.
  - We also better explain that the objective of the paper is not to use thermodynamics to search for a storage optimum but to propose a useful perspective to discriminate differences in storage dynamics that reflect a different interplay of capillary and gravitational controls.
- Section 2 (Theoretical background) has been streamlined as recommended by Gerrit de Rooij:
  - We largely follow the avenue of Bolt and Frissel (1960) when expressing the matric and the gravity potentials by their energetic counterparts.
  - Yet we stick to our guns with respect to the chosen terminology. This has good reasons and we clearly state those in this section.
- Section 3 (Application):
  - We followed Gerrits de Rooijs idea and compiled the energy state functions for the bin centroids of the histogram of HAND in the study areas, and provide both signatures for the two catchments.
  - We better explained the choice of our representative soil water retention function.
- Section 4 (Results) is largely unchanged:
  - Except that we added two figures that reveal in both areas a threshold-like relation between the observed free energy of soil water in the riparian zone and observed streamflow. Note that the tipping points coincide with the local equilibrium state of zero free energy, which suggests that emergence of a potential energy excess/storage excess in the riparian zone coincides with the onset of storage controlled direct streamflow generation. While such threshold behavior is not unusual, it is remarkable that the tipping is consistent with the underlying theoretical basis.
- Section 5: Has been streamlined and we completely removed section 5.3 of the original manuscript.

I hope you forgive that I will did not copy our detailed replies to both reviewers in the annex, as these are provided in the discussion.

Best regards,

Erwin Zehe

[revised manuscript text omitted]

---

## Referee Report (RR1)

Review of the revision of hess-2018-346

Gerrit de Rooij.
December 2018

Due to time constraints I almost exclusively focused on the revised section of the paper, so I may have overlooked some things and therefore request clarifications that are already there. The paper improved notably, but I find some things in the new theoretical section that I believe warrant attention. I explain these in detail (and offer an alternative equation and its discussion) below.

l. 24: 'Both study areas…' At this point you have not mentioned any study area yet.

l. 44-45: This is the special case of no flow, which is approximated in some locations during limited periods of time. Steady-state conditions also arise when there is flow, as long as the flow field does not change. Unit gradient flow is an example of such a steady flow, and one that is probably more abundant than hydrostatic equilibrium, because it tends to occur below the root zone in deep vadose zones. In semi-arid areas there is some evidence (although I cannot produce references) of unit gradient conditions prevailing for many years.

In the wording of the line of argument you develop in l. 48-50, unit gradient flow is a condition in which the capillary gradient has vanished, and the flow is purely controlled by gravity. This does not mean the influence of capillarity can be neglected. On the contrary, the capillary forces determine the water content and the matric potential under which the unit gradient condition will arise, given the steady water flux that results from the long-term average of the net infiltration. I would therefore argue that unit gradient flow presents a case where the interplay between gravity and capillarity has found a balance, and therefore can be invoked to support your argument.

l. 54: I find geology to be the odd one out. Tectonic uplift keeps up with erosion, or even outruns it. Therefore, the existence of mountain ranges or smaller geological features is not really co-evolutionary. Pedogenesis and geomorphology (including incision of rivers, mudslides, and everything else that is driven by (partially subsurface) water), etc. therefore are processes that respond to the geological drivers. I do not see any feedback into the geological processes that the term co-evolution suggests.

Another issue is the difference in time scales of geology and the other processes. Many of the lower mountain ranges in the world (Ural, Appalachians, German Mittelgebirge) are remnants of the large mountain range of Gondwanaland. It is difficult to argue that these ranges co-evolve with soils that developed in the Holocene. A more hydrological example of lack of co-evolution is provided by the aquifers below the Sahara and the Saudi-Arabian desert that are remnants of less extreme climatic conditions and have little or no bearing on the processes in the top meters of the subsurface. This water is fossil, and is therefore sometimes considered geological in nature, unconnected as it is from the current hydrological cycle, were it not for anthropogenic interference through pumping.

l. 113: New paragraph?

l. 119-128:  I like this paragraph - it clearly outlines what we can and cannot expect when we travel down the thermodynamical avenue.

l. 135-136.  ...energy is an additive quantity, while .. gravity and matric potentials are not.
This needs some clarification: we can and do add the gravitational and the matric potential all the time.

l. 143: ...optimizes...infiltration, moisture retention and drainage of catchments.

The question how and through what mechanism this optimum is defined. In more plain terms: what is a catchments' objective function, and how did the catchment find it?

But you do not need to go into that in this paper.

l. 169: The volume $V$ for which you define the Gibbs free energy contains three phases, but you leave out the gas phase. In Eq. (2), the work term can be set to zero for the solid and the liquid phase because they can be considered incompressible. But there will be a non-zero contribution to this term for the gas phase.

Perhaps it is easier to formally limit the analysis to the water phase only of your control volume. The work term in Eq. (2) is then multiplied by theta and subsequently declared to be zero because water can be considered incompressible for normally encountered pressure ranges.

Below Eq. (1) and possibly elsewhere there are many inconsistencies in the math. The notation of units sometimes uses the division symbol ('/'), sometimes negative powers. The same variable appears in normal and in italic font, or even in upper and lower case (Gibbs free energy). This is confusing.

l. 175: The constant $g$ does not denote the acceleration of the planet itself, but that generated by its gravitational field.

l. 195-201 (Discussion of the meaning of potential): I belong to the category of people that were taught to use the term potential for every quantity of which the gradient drives a flux of some sort, and for which the flux is proportional to that gradient. The resulting flow is termed a potential-driven flow. If the potential is a voltage, Ohm's Law emerges. If it is a temperature, Fourier's Law of heat conduction arises. In case of a solute concentration, Fick's Law of diffusion appears. If we have a hydraulic potential, we arrive at Darcy's Law.

Within the Darcian framework, the potential energies derived from the position in the gravitational field, from the position in the pore architecture that gives rise to the matric forces, from the osmotic potential derived from the presence of solutes, etc. are all considered energies that are additive. They all can perform work, so the term potential energy seems justified. The gravitational field stands out because it is independent of any property of the soil system, such as the amount of water present in the soil, the architecture of the pore space, etc. The other force fields are dynamic and influenced by the system of which the water upon which they operate is a constituent, but they are considered to exert a force on the water, just like gravity.

That being said, feel free to keep this explanation in the text. This allows both positions to be debated in the literature in the open, which is the proper way.

Eq. (4): I think you should use the general version, with the two principal radii of curvature. The limitation to cylindrical pores is neither desirable nor necessary here.

Eqs. (3) and (5): In Eq. (3) you only considered a change in the matric potential and gravitational potential energies, while keeping the water content constant. The terms $dp$ and $dz$ reflect infinitesimal changes in the matric potential and position in the gravitational field. The term $dz$ is intuitively clear. The term $dp$ is harder. The only way I can think of changing the matric potential in an isothermal system where the properties of the liquid and the solid phase do not change is through the pressure of the gas phase without an equal change in the atmospheric pressure. Thermodynamically, the term with $dp$ in Eq. (3) makes sense, but it is not so easy to find a physical mechanism to create the infinitesimal change at constant water content.

In Equation (5) you keep the position $z$ constant and do not permit a change in the matric potential energy when neither the water content nor the position is changed. Equation (5) therefore is not the derivative of Eq. (3), contrary to what the paper states. Instead, by replacing $dz$ and $dp$ by $z$ and $\psi$ you do not permit the position of the water in the gravitational field and its elusive equivalent for the matric potential field to change. Instead you change the water content by an infinitesimal amount and show how it affects the potential energy of the water. This is much easier to interpret that the term $dp$ in Eq. (3) because it is immediately obvious that the matric potential changes with the water content. The change in gravitational potential energy is also clear.

I believe the first term of the right-hand-side of Eq. (5) is incorrect though. Let me explain by carrying out the derivation operation on amounts of potential energy stored in a volume of water that is subjected to a small change. The volume of water is that in an arbitrary volume $V$ with volumetric water content $\theta$. Without loss of generality, the units of $V$ are chosen such that the volume of water within it constitutes one arbitrary unit of volume. The amount of matric energy in that volume is than equal to $\rho g \psi \theta$, and the amount of gravitational potential energy equals $\rho g z \theta$ (kg m s$^{-2}$), consistent with your notation. We now add an infinitesimal amount of water $d\theta$ in an infinitesimal time interval $dt$ and calculate the resulting change in both potential energies:

$$\rho g \frac{\partial(\psi\theta)}{\partial t} + \rho g \frac{\partial(z\theta)}{\partial t} = \rho g \left( \psi \frac{\partial \theta}{\partial t} + \theta \frac{\partial \psi}{\partial t} + z \frac{\partial \theta}{\partial t} + \theta \frac{\partial z}{\partial t} \right) = \rho g \left( \psi \frac{\partial \theta}{\partial t} + \theta \frac{d\psi}{d\theta} \frac{\partial \theta}{\partial t} + z \frac{\partial \theta}{\partial t} \right)$$

You omitted the first term within the parentheses, which represents the change in the amount of matric potential energy caused by the change in the amount of water. Note also that the use of partial derivatives in the $d\psi/d\theta$ is incorrect because you assume that the function $\psi(\theta)$ completely describes the behavior of $\psi$. This assumption is necessary and sufficient to allow $\partial\psi/\partial t$ to be expressed as the product of $d\psi/d\theta$ and $\partial\theta/\partial t$, so this is not trivial.

With this expression you can nuance the statement in l. 218-220. Only when $d\psi/d\theta = 0$ (under saturation or complete dryness, the latter being of little interest in a hydrological journal) are the changes in matric and gravitational potential energy equal (and of the

same sign). With $d\psi/d\theta > 0$ in hydrophilic soils, the product $\theta d\psi/d\theta$ determines by what additional amount the potential energy in $V$ changes when $\theta$ is changed by an amount $d\theta$.

The statement in l. 221-223 is incorrect in two ways. The sum $(\psi + z)$ denotes the potential energy of the water around a point where the values of the matric potential and $z$ are as indicated, but only there. Multiplying this sum by the water content denotes the potential energy contained by the water in a small volume element of bulk soil surrounding that point. For larger volumes, you need the volume integral of this sum. This amounts to the sum of Eqs. (8) and (9) in de Rooij (2009), but without the division by the total amount of water in the integration volume. Section 2.3 of de Rooij (2011) addresses this issue more thoroughly by presenting formal definitions that conserve energy during upscaling operations, and also explains why the intrinsic phase average is more elegant than the phase average. Because Eq. (5) refers to a single point, this is not relevant at this point, but as soon the analysis moves to larger scales this aspect becomes important.

l. 239-241: The equation is valid irrespective of the value of the integration constant, which only reflects the reference height for the vertical coordinate. You are working in catchments with varying groundwater levels in space and time, so I do not think it is wise to fix the reference height to the groundwater level at an arbitrary point and an arbitrary time, which is what you do when you fix it to 'the groundwater level'. If you really like the Gibbs free energy to go to zero it is more correct to state that you set $c$ to zero without loss of generality because it reflects the vertical position with respect to an arbitrary datum.

l. 242-243: I recommend to include a remark that this approach assumes hydrostatic equilibrium with a fixed groundwater level in the entire unsaturated zone, for the non-soil physicists that read HESS.

l. 264-265. I associate storage with a certain depth interval (e.g., the entire unsaturated zone). That would be equal to the integral of the water content over that depth interval. But here you use it for the water content at the top of the interval only. Why?

l. 277-278. Not only do you assume capillary contact with the groundwater, but you assume hydrostatic equilibrium throughout the profile. Capillary contact with the groundwater will be there as long as the water does not retract into pendular rings. It will simply not play much of a role higher up in the profile. This makes the assumption of hydrostatic equilibrium a rather strong one.

Figure 2. The term 'water stock' is definitely misleading here. You only concern yourself with the water in a plane at a given height above the water table, not the water below and above that plane. For that you need volume integrals. See de Rooij (2011) for the underlying theory, including the effect of the geometry of the volume of interest.

At degrees of saturation of about 0.05 (Colpach), 0.35 (Weiherbach) and 0.72 (Wollefsback) the gravitational potential contributes about 1% to the total free energy for the chosen depth to groundwater, judging from Fig. 1. From there on, Fig. 2 basically is the retention curve with the logarithmic axis replaced by a linear axis. What worries me about this figure reflects what worries me about Fig. 5: the changing amount of

water with changing matric potential is not taken into account at all. From a catchment-scale perspective this is really dangerous – you cannot really tell much about the energy status of the catchment water if you do not weigh the local energies with the local water contents. We are back to the proper way to carry out volume integrations again. This plays into the discussion at line 300, where you use the term energy deficit. But you cannot quantify this correctly because you are only able to determine the deficit of potential energy per volume (or weight) of water without being able to see the difference in volumes water at the current non-equilibrium state and the equilibrium state. But we can do that already with the pF curve, we do not need the free energy for that.

l. 301. You use the term 'dry cohesive soils'. Why does the soil need to be cohesive for the rapid deviation from equilibrium to occur? Also, you discuss excursions away from and back to equilibrium. The soil cannot be that dry away from equilibrium, can it? Or are you talking of sands at pF 3 (consistent with 10 m deep groundwater)? In that case, the assumption of hydrostatic equilibrium with the groundwater table is illusionary. More generally, you can argue that the sensitivity to perturbations is determined by $d\psi/d\theta$, which typically is very large near saturation and in the dry end.

l. 304-305. The grammar of this sentence seems to be wrong, or perhaps there is a devilish typo.

l. 304-311. The soil's behaviour will vary dramatically with the chosen reference matric potential (because that is what you fix when you set a depth to groundwater in combination with the requirement of hydrostatic equilibrium).

l. 318-319. This is the case for draining rivers. For rivers that lose water, c is larger than 1. This becomes relevant when there are ditches instead of rivers and water is let in during summer to water the soils.

l. 383: the pF requires the absolute value of $\psi$.

l. 388: To arrive at the stored water amount in a landscape for a given tension you need to multiply the average water content by the volume of the portion of the landscape to which the tension applies.

l. 413: a pore space of less than 20 meters? I do not understand.

Figure 5. Are the energy stare functions based on a single groundwater depth again? If so, what was this depth, how was it selected, and how representative are these curves for the catchment in view of my comments above?

I do not fully grasp the vertical scales of the figures. Panel b shows that Colpach has depths to drainage between 2 and 56 m or so (I can hardly see the tick marks of the graph). You plotted the free energy between -10 and 30 m, so there seem to be about 15 m of the total range missing. If the range of panel a is more or less centered on the range of HAND values, this would lead to a reference depth to groundwater of roughly 20 m (when the range in panel a covers 10 to 50 m of HAND values). This does not seem to be representative at all of the distribution of HAND in the catchment.

For Wollefsbach, HAND ranges from 1 to 33 m, yet the range of the free energy is 80 meters. I have no idea how to interpret this or speculate about the chosen reference depth to groundwater.

Figure 6. Does the range of free energy in Wollefsbach reflect considerable drying in summer? Does that not invalidate your assumption of equilibrium with a ground water table that cannot be that deep according the reported HAND values for that catchment? Because the curve of the free energy against degree of saturation increasingly resembles the non-logarithmic retention curve I can imagine this does not matter too much, but it should perhaps be discussed.

You report alternative values of the depth to groundwater, so I assume I overlooked the best values (I only reviewed the changes, because of time constraints). Do you explain how you found these? Neither value for Colpach seems to match the HAND histogram in Fig. 5, and you do not indicate the values for Wollefsbach.

l. 531 ... large values of HAND...?

l. 686-688: I did not see much evidence for a linear dependence of the free energy on the degree of saturation (nor did I expect it). Please elaborate.

I would like to have some clarification on the determination of the depth to groundwater that separates the wet and the dry branches of your curves.

References

de Rooij, G.H. 2009. Averaging hydraulic head, pressure head, and gravitational head in subsurface hydrology, and implications for averaged fluxes, and hydraulic conductivity. Hydrol. Earth Syst. Sci. 13:1123-1132. www.hydrol-earth-syst-sci.net/13/1123/2009/

de Rooij, G.H. 2011. Averaged water potentials in soil water and groundwater, and their connection to menisci in soil pores, field-scale flow phenomena, and simple groundwater flows. Hydrol. Earth Syst. Sci. 15:1601-1614. doi:10.5194/hess-15-1601-2011

---

## Author Response (AR2)

Reply to re-review of Gerrit de Rooij (GR)
January 2019

GR: Due to time constraints I almost exclusively focused on the revised section of the paper, so I may have overlooked some things and therefore request clarifications that are already there. The paper improved notably, but I find some things in the new theoretical section that I believe warrant attention. I explain these in detail (and offer an alternative equation and its discussion) below.

EZ: I read Gerrit de Rooij's assessment of our revised manuscript with great interest and pleasure and thank him again for his efforts and thorough reflection. Please find my detailed answers below.

GR: l. 24: 'Both study areas…' At this point you have not mentioned any study area yet.
EZ: This is reformulated in the revised manuscript.

GR: l. 44-45: This is the special case of no flow, which is approximated in some locations during limited periods of time. Steady-state conditions also arise when there is flow, as long as the flow field does not change. Unit gradient flow is an example of such a steady flow, and one that is probably more abundant than hydrostatic equilibrium, because it tends to occur below the root zone in deep vadose zones. In semi-arid areas there is some evidence (although I cannot produce references) of unit gradient conditions prevailing for many years.

In the wording of the line of argument you develop in l. 48-50, unit gradient flow is a condition in which the capillary gradient has vanished, and the flow is purely controlled by gravity. This does not mean the influence of capillarity can be neglected. On the contrary, the capillary forces determine the water content and the matric potential under which the unit gradient condition will arise, given the steady water flux that results from the long-term average of the net infiltration. I would therefore argue that unit gradient flow presents a case where the interplay between gravity and capillarity has found a balance, and therefore can be invoked to support your argument.

EZ: This statement is indeed unprecise as a steady state does not necessarily correspond to hydraulic equilibrium conditions. We thus change the statement to: "Steady state, hydraulic equilibrium conditions imply"….

l. 54: I find geology to be the odd one out. Tectonic uplift keeps up with erosion, or even outruns it. Therefore, the existence of mountain ranges or smaller geological features is not really co-evolutionary. Pedogenesis and geomorphology (including incision of rivers, mudslides, and everything else that is driven by (partially subsurface) water), etc. therefore are processes that respond to the geological drivers. I do not see any feedback into the geological processes that the term co-evolution suggests.  Another issue is the difference in time scales of geology and the other processes. Many of the lower mountain ranges in the world (Ural, Appalachians, German Mittelgebirge) are remnants of the large mountain range of Gondwanaland. It is difficult to argue that these ranges co-evolve with soils that developed in the Holocene. A more hydrological example of lack of co-evolution is provided by the aquifers below the Sahara and the Saudi-Arabian desert that are remnants of less extreme climatic conditions and have little or no bearing on the processes in the top meters of the subsurface. This water is fossil, and is therefore sometimes considered geological in

nature, unconnected as it is from the current hydrological cycle, were it not for anthropogenic interference through pumping.

E Z. Good point! What we in fact mean is that the geological setting constrains the evolution of the soil material. We changed the wording accordingly as follows: 'The climatological and geological setting constrains the co-development or co-evolution of soils, geomorphology and vegetation (as suggested by e.g. Troch et al., 2015; Sivapalan and Bloschl, 2015; Saco and Moreno-de las Heras, 2013). One might hence wonder whether this constrained co-development created a distinctly typical interplay of capillary and gravitational controls on soil moisture'

GR: l. 113: New paragraph?
EZ: done

l. 119-128: I like this paragraph - it clearly outlines what we can and cannot expect when we travel down the thermodynamical avenue.
EZ: Thank you for this nice comment.

GR: l. 135-136. ...energy is an additive quantity, while .. gravity and matric potentials are not. This needs some clarification: we can and do add the gravitational and the matric potential all the time.

EZ: Agreed! In precise terms energy is an extensive quantity/state variable (such as mass, momentum, entropy, electrical charge), while potentials are intensive state variables (such as temperature, pressure, velocity, chemical potential). Extensive state variable are discontinuous an interfaces and they grow in an additive manner when the volume of a system is enlarged or the two systems are merged. Extensive variables are thus stock variables that can be balanced. Intensive state variables are continuous at interfaces and are not additive in the above explain sense. If we open the door between two rooms of the same temperature, temperatures don't add up (thermal energy does though). We change the wording as follows: "Secondly, energy is an extensive quantity, as such it is additive when different systems are merged, it grows with increasing system size and changes can be described through a balance. One may hence apply volumetric averaging and upscaling to energy for instance to derive macroscale effective constitutive relations and macroscale equations as shown by de Rooij (2009, 2011). In contrary the related gravity and matric potentials are intensive state variables and as such neither additive in the above specified sense, nor can their changes be balanced."

GR: l. 143: ...optimizes...infiltration, moisture retention and drainage of catchments. The question how and through what mechanism this optimum is defined. In more plain terms: what is a catchments' objective function, and how did the catchment find it? But you do not need to go into that in this paper.

EZ: I absolutely agree that this is the non-trivial to define such an optimum. The system could reach such an optimum through "mutation and selection". So this is not a target process but it implies that landscapes which do not develop are less resilient against disturbance as those how are in an optimum.

GR l. 169: The volume V for which you define the Gibbs free energy contains three phases, but you leave out the gas phase. In Eq. (2), the work term can be set to zero for the solid and the liquid phase because they can be considered incompressible. But there will be a non-zero contribution to this term for the gas phase.

Perhaps it is easier to formally limit the analysis to the water phase only of your control volume. The work term in Eq. (2) is then multiplied by theta and subsequently declared to be zero because water can be considered incompressible for normally encountered pressure ranges.

EZ: I agree that the work term cannot be neglected for the air phase. And please note that we state that we neglect the work term, because we focus on free energy of the water phase, right below equation 2. In the revised manuscript we additionally state this in brackets

GR: Below Eq. (1) and possibly elsewhere there are many inconsistencies in the math. The notation of units sometimes uses the division symbol ('/'), sometimes negative powers. The same variable appears in normal and in italic font, or even in upper and lower case (Gibbs free energy). This is confusing.

EZ. I apologize for this and we fixed this in the revised manuscript.

l. 175: The constant g does not denote the acceleration of the planet itself, but that generated by its gravitational field.

EZ. I apologize for the denglish. The german term is "Erdbeschleunigung" and I just did a word for word translation. We changed it to gravitational acceleration

GR: l. 195-201 (Discussion of the meaning of potential): I belong to the category of people that were taught to use the term potential for every quantity of which the gradient drives a flux of some sort, and for which the flux is proportional to that gradient. The resulting flow is termed a potential-driven flow. If the potential is a voltage, Ohm's Law emerges. If it is a temperature, Fourier's Law of heat conduction arises. In case of a solute concentration, Fick's Law of diffusion appears. If we have a hydraulic potential, we arrive at Darcy's Law. Within the Darcian framework, the potential energies derived from the position in the gravitational field, from the position in the pore architecture that gives rise to the matric forces, from the osmotic potential derived from the presence of solutes, etc. are all considered energies that are additive. They all can perform work, so the term potential energy seems justified. The gravitational field stands out because it is independent of any property of the soil system, such as the amount of water present in the soil, the architecture of the pore space, etc. The other force fields are dynamic and influenced by the system of which the water upon which they operate is a constituent, but they are considered to exert a force on the water, just like gravity.

That being said, feel free to keep this explanation in the text. This allows both positions to be debated in the literature in the open, which is the proper way.

EZ: Good point and fair enough. I was taught that potential energy relates to the position of a test body in an elementary force field: either a test mass in the gravity field or a test electrical charge in the electrostatic field. Please note that we use the term chemical energy although they relate to the product of the chemical potential and the mass.

GR: Eq. (4): I think you should use the general version, with the two principal radii of curvature. The limitation to cylindrical pores is neither desirable nor necessary here.

EZ: done.

GR: Eqs. (3) and (5): In Eq. (3) you only considered a change in the matric potential and gravitational potential energies, while keeping the water content constant. The terms dp and dz reflect infinitesimal changes in the matric potential and position in the gravitational field. The term dz is intuitively clear. The term dp is harder. The only way I can think of changing the matric potential in an isothermal system where the properties of the liquid and the solid phase do not change is through the pressure of the gas phase without an equal change in the atmospheric pressure. Thermodynamically, the term with dp in Eq. (3) makes sense, but it is not so easy to find a physical mechanism to create the infinitesimal change at constant water content.

In Equation (5) you keep the position z constant and do not permit a change in the matric potential energy when neither the water content nor the position is changed.

EZ: The confusion is because of the fact that we have been moving away from the notations of the old masters. Neither equation 1 nor equation 3 is a total differential in the mathematical sense (I remember we stated this in a foot note in the previous manuscript, which dissipated during the revision). ==This is why classical text books on thermodynamics use the symbol δ instead of the d, and speak of a variation, which implies that the other factors in the product remain constant. (We added this note to the text).==

$$\delta g_{\text{free}} = \rho g \theta \delta \psi + \rho g \theta \delta z = \frac{\partial g_{\text{free}}}{\partial \psi} d\psi + \frac{\partial g_{\text{free}}}{\partial z} dz$$

The fact that this is possible is usually attributed to the fact that the system is in contact with "reservoirs". The most classical example is the heat reservoir, which may release thermal energy without changing its temperature. Honestly, I had always problems in imagining this as student. And it becomes more difficult to imagine when moving away from technical systems to natural system, as fluxes feedback on their driving boundary conditions and "the reservoir idea" becomes in most cases meaningless. In case you are interested, you may look at the my discussion of Zehe et al (2013) in HESS. In the revised manuscript we now state that Equations 1 and 3 are not total differentials.

GR: Equation (5) therefore is not the derivative of Eq. (3), contrary to what the paper states. Instead, by replacing dz and dp by z and ψ you do not permit the position of the water in the gravitational field and its elusive equivalent for the matric potential field to change. Instead you change the water content by an infinitesimal amount and show how it affects the potential energy of the water. This is much easier to interpret that the term dp in Eq. (3) because it is immediately obvious that the matric potential changes with the water content. The change in gravitational potential energy is also clear.

I believe the first term of the right-hand-side of Eq. (5) is incorrect though. Let me explain by carrying out the derivation operation on amounts of potential energy stored in a volume of water that is subjected to a small change. The volume of water is that in an arbitrary volume V with volumetric water content θ. Without loss of generality, the units of V are chosen such that the volume of water within it constitutes one arbitrary unit of volume. The amount of matric energy in that volume is than equal to $\rho g \psi \theta$, and the amount of gravitational potential energy equals $\rho g z \theta$ (kg m s-2), consistent with your notation. We now add an infinitesimal amount of water dθ in an infinitesimal time interval dt and calculate the resulting change in both potential energies:

$$\rho g \frac{\partial(\psi\theta)}{\partial t} + \rho g \frac{\partial(z\theta)}{\partial t} = \rho g \left( \psi \frac{\partial\theta}{\partial t} + \theta \frac{\partial\psi}{\partial t} + z \frac{\partial\theta}{\partial t} + \theta \frac{\partial z}{\partial t} \right) = \rho g \left( \psi \frac{\partial\theta}{\partial t} + \theta \frac{d\psi}{d\theta} \frac{\partial\theta}{\partial t} + z \frac{\partial\theta}{\partial t} \right)$$

You omitted the first term within the parentheses, which represents the change in the amount of matric potential energy caused by the change in the amount of water. Note also that the use of partial derivatives in the $d\psi/d\theta$ is incorrect because you assume that the function $\psi(\theta)$ completely describes the behavior of $\psi$. This assumption is necessary and sufficient to allow $\partial\psi/\partial t$ to be expressed as the product of $d\psi/d\theta$ and $\partial \psi /\partial t$, so this is not trivial.

EZ: Touché! To save our honor, we used the correct version proposed by GR in the study of Zehe et al (2013), in the psi based form though (Equation 7 in Zehe et al. 2013). We changed equation 5 in our manuscript accordingly. But note that this is generally not as clear as one might think. We come to the equation proposed by GR if the equation $g_{free} = \rho g\theta\psi + \rho g\theta z$ is correct, and we do believe that this is the case. But this must not be the case for all forms of energy.

What I learnt about the Gibbs fundamental theorem is that we generally **cannot** quantify the total energy content of a system by summing up all the energies, but that we can only quantify the changes in such an additive way as described by this form. This is because the total energy amount can only be quantified for a few energy forms but not for all (through their Gibbs function). We can do this for potential energy, kinetic energy, electrical energy. We can't do this for surface energy, chemical energy and or heat. We express the change in heat/thermal energy as dQ=TdS ($\delta$Q=T$\delta$S), but Q=TS is wrong! We may express the change in chemical energy $\delta$ E=$\mu$ $\delta$ M is correct but E= $\mu$ $\delta$ M etc.

GR: With this expression you can nuance the statement in l. 218-220. Only when $d\psi/d\theta = 0$ (under saturation or complete dryness, the latter being of little interest in a hydrological journal) are the changes in matric and gravitational potential energy equal (and of the same sign). With $d\psi/d\ d\theta > 0$ in hydrophilic soils, the product $\theta\ d\psi/d\theta$ determines by what additional amount the potential energy in V changes when $\theta$ is changed by an amount d$\theta$.

The statement in l. 221-223 is incorrect in two ways. The sum ($\psi$ + z) denotes the potential energy of the water around a point where the values of the matric potential and z are as indicated, but only there. Multiplying this sum by the water content denotes the potential energy contained by the water in a small volume element of bulk soil surrounding that point. For larger volumes, you need the volume integral of this sum. This amounts to the sum of Eqs. (8) and (9) in de Rooij (2009), but without the division by the total amount of water in the integration volume. Section 2.3 of de Rooij (2011) addresses this issue more thoroughly by presenting formal definitions that conserve energy during upscaling operations, and also explains why the intrinsic phase average is more elegant than the phase average. Because Eq. (5) refers to a single point, this is not relevant at this point, but as soon the analysis moves to larger scales this aspect becomes important.

EZ: ($\psi$ + z) is the free energy density in the water phase at a point or in the wetted part of an infinitesimally small volume consisting of water only. $\theta(\psi$ + z) is the free energy density normalized with the entire volume. I think we stated this clearly and we stress that this is the free energy density at elevation z. For larger volumes you need to integrate $\theta$ ($\psi$ + z), because the volume is only partly occupied by water. Otherwise this violates energy conservation as correctly stated by de Rooij (2009).

We changed the entire passage GR is referring to as follows:

$$\frac{\partial g_{\text{free}}}{\partial t} = \frac{\partial(e_{\text{pot}} + e_{\text{cap}})}{\partial t} = \rho g \left[ (\psi + \theta \frac{d\psi}{d\theta}) \frac{\partial \theta}{\partial t} + z \frac{\partial \theta}{\partial t} \right] \text{ Eq. (5).}$$

Note that the potential energy density of soil water (the second term on the right hand side) increases linearly with increasing soil water content. On the other hand, capillary binding energy decreases with increasing soil water content, as the absolute value of the matric potential declines non-linearly with increasing soil water content. The change in capillary energy density with a given change in soil water content is determined by the product of the actual soil water and the slope of the water retention curve. We thus state that the product of the well-known soil hydraulic potential and the soil water content corresponds to the volumetric density of free energy of soil water per unit weight. The free energy of soil water for a larger volume is the volume integral of the total hydraulic potential times the soil water content over the volume of interest (de Rooij, 2009; Zehe et al., 2013):

$$E_{\text{free}} = E_{\text{cap}} + E_{\text{pot}} = \int \rho g (\psi(\theta) + z) \theta dV \text{ Eq. (6)}$$

The latter reflects both the binding state and the amount of water stored in a control volume above groundwater and thus reflects the local retention properties and the topographic setting as well. Note that the change in potential energy of soil water at a given elevation scales linearly with the soil water content. One might thus wonder whether the dominance of the one or the other energy form may at least partly influence whether a system behaves in a linear or non-linear fashion.

GR: 239-241: The equation is valid irrespective of the value of the integration constant, which only reflects the reference height for the vertical coordinate. You are working in catchments with varying groundwater levels in space and time, so I do not think it is wise to fix the reference height to the groundwater level at an arbitrary point and an arbitrary time, which is what you do when you fix it to 'the groundwater level'. If you really like the Gibbs free energy to go to zero it is more correct to state that you set c to zero without loss of generality because it reflects the vertical position with respect to an arbitrary datum.

EZ: I learned that an indefinite integration adds a constant to antiderivative and that this constant is determined at the system boundary. I do not see any reasons for setting c to zero expect the one that the matric potential becomes zero at the groundwater surface. This is now stated in the text.

GR: l. 242-243: I recommend to include a remark that this approach assumes hydrostatic equilibrium with a fixed groundwater level in the entire unsaturated zone, for the non-soil physicists that read HESS.

EZ: To be precise, we assume hydraulic equilibrium within the entire saturated zone. This is stated. The groundwater level does not need to be fixed, a declining or rising GW level will change the equilibrium profile.

GR: l. 264-265. I associate storage with a certain depth interval (e.g., the entire unsaturated zone). That would be equal to the integral of the water content over that depth interval. But here you use it for the water content at the top of the interval only. Why?

EZ. I removed storage here.

GR: l. 277-278. Not only do you assume capillary contact with the groundwater, but you assume hydrostatic equilibrium throughout the profile. Capillary contact with the groundwater will be there as long as the water does not retract into pendular rings. It will simply not play much of a role higher up in the profile. This makes the assumption of hydrostatic equilibrium a rather strong one.

EZ: Correct and this is clearly stated at the beginning of the section.

GR: Figure 2. The term 'water stock' is definitely misleading here. You only concern yourself with the water in a plane at a given height above the water table, not the water below and above that plane. For that you need volume integrals. See de Rooij (2011) for the underlying theory, including the effect of the geometry of the volume of interest. At degrees of saturation of about 0.05 (Colpach), 0.35 (Weiherbach) and 0.72 (Wollefsback) the gravitational potential contributes about 1% to the total free energy for the chosen depth to groundwater, judging from Fig. 1. From there on, Fig. 2 basically is the retention curve with the logarithmic axis replaced by a linear axis.

EZ: I agree with GR that figure 2 relates pretty strong to shape of the retention curve. Yet it is different as $e_{free} = \theta (\psi + z)$ contains as a term the product of the soil water content and the matric potential. This creates a different shape, compared to the plot of the hydraulic potential, which would be for the Colpach pretty horizontal for larger saturations and then follow the retention curve. And we omitted the term water stock as recommended.

GR: What worries me about this figure reflects what worries me about Fig. 5: the changing amount of water with changing matric potential is not taken into account at all. From a catchment-scale perspective this is really dangerous – you cannot really tell much about the energy status of the catchment water if you do not weight the local energies with the local water contents. We are back to the proper way to carry out volume integrations again. This plays into the discussion at line 300, where you use the term energy deficit. But you cannot quantify this correctly because you are only able to determine the deficit of potential energy per volume (or weight) of water without being able to see the difference in volumes water at the current non-equilibrium state and the equilibrium state. But we can do that already with the pF curve, we do not need the free energy for that.

EZ: Sorry that I disagree. The curve accounts for the changing water content as it is the product of the $\theta (\psi + z)$. In Zehe et al. (2013) we analysed in fact the volume integrated values of free energy (taken from a calibrated model). This can be helpful, but with the integration we lose information about the distribution of energy within the system (similar as we lose information about the distribution of potentials, if we work with integral averages). Figure 5 provides information about the stratification of the energy along a representative distribution of geo-potential levels in the catchment. This is much more than an integral can provide. And as already shown in our reply to GR's last review, the information can of course be integrated (when using a calibrated model).

GR: l. 301. You use the term 'dry cohesive soils'. Why does the soil need to be cohesive for the rapid deviation from equilibrium to occur? Also, you discuss excursions away from and back to equilibrium. The soil cannot be that dry away from equilibrium, can it? Or are you talking of sands at pF 3 (consistent with 10 m deep groundwater)? In that case, the assumption of hydrostatic equilibrium with the groundwater table is illusionary. More

generally, you can argue that the sensitivity to perturbations is determined by dψ/dθ , which typically is very large near saturation and in the dry end.

EZ: This is obviously not well phrased. We wanted to express that small changes in the soil water content during dry conditions may cause large changes in the energy state, dψ/dθ. We clarified this in the manuscript.

GR: l. 304-305. The grammar of this sentence seems to be wrong, or perhaps there is a devilish typo.

EZ. We changed this to: Figure 2 shows that the three different soils at the same geopotential level, are characterized by distinctly different energy state curves as function of relative saturation.

GR: l. 304-311. The soil's behaviour will vary dramatically with the chosen reference matric potential (because that is what you fix when you set a depth to groundwater in combination with the requirement of hydrostatic equilibrium).
EZ: Absolutely true, we elaborate on this in detail in the discussion of the paper. Personally I think this is an advantage that the energy state curve is sensitive to depth to groundwater. This allows an estimation of the ground water level based in available pairs of soil water content, matric potential data and the local retention function, as we intend to show in a forthcoming paper.

GR l. 318-319. This is the case for draining rivers. For rivers that lose water, c is larger than 1. This becomes relevant when there are ditches instead of rivers and water is let in during summer to water the soils.
EZ: Good point, we stress this in the revised manuscript.

GR: l. 383: the pF requires the absolute value of ψ.
EZ: We corrected this.

GR l. 388: To arrive at the stored water amount in a landscape for a given tension you need to multiply the average water content by the volume of the portion of the landscape to which the tension applies.

EZ: What we meant is "The averaged soil water content at each matric potential/tension-level $\bar\theta(\psi)$ is an estimator to the expectation value of the soil water content at this tension"

GR: l. 413: a pore space of less than 20 meters? I do not understand.
EZ. This is a misunderstanding. We mean. "The absolute values of $e_{free}$ are in the corresponding C-regime less than 20m"

GR: Figure 5. Are the energy stare functions based on a single groundwater depth again? If so, what was this depth, how was it selected, and how representative are these curves for the catchment in view of my comments above?
I do not fully grasp the vertical scales of the figures. Panel b shows that Colpach has depths to drainage between 2 and 56 m or so (I can hardly see the tick marks of the graph). You plotted the free energy between -10 and 30 m, so there seem to be about 15 m of the total range missing. If the range of panel a is more or less centered on the range of HAND values, this would lead to a reference depth to groundwater of roughly 20 m (when the range in panel a covers 10 to 50 m of HAND values). This does not seem to be representative at all of

the distribution of HAND in the catchment.

For Wollefsbach, HAND ranges from 1 to 33 m, yet the range of the free energy is 80 meters. I have no idea how to interpret this or speculate about the chosen reference depth to groundwater.

EZ.: We thought that section 2. made clear that we use HAND as an estimator of depth to ground water. To better stress this we added the following statement to the beginning of section "Note that we use HAND as an estimator for the depth to groundwater here."

That said it becomes clear that the free energy is at a saturation of 1 not equal to the HAND but equal to the product of Hand and the soil water content at saturation. We added to the figure caption: "Please note that $e_{free}$ at a relative saturation of 1 equals the product of HAND and the soil water content at saturation." to avoid similar confusion.

GR: Figure 6. Does the range of free energy in Wollefsbach reflect considerable drying in summer? Does that not invalidate your assumption of equilibrium with a ground water table that cannot be that deep according the reported HAND values for that catchment? Because the curve of the free energy against degree of saturation increasingly resembles the non-logarithmic retention curve I can imagine this does not matter too much, but it should perhaps be discussed.

EZ: We do not assume an equilibrium with ground water. We just assume this for the definition of the equilibrium point. What we in fact see is that the system deviates rather far from this equilibrium but it also relaxes back to it. This is by the way our main hypothesis stated in at the end to the introduction, and it is nicely corroborated.

GR: You report alternative values of the depth to groundwater, so I assume I overlooked the best values (I only reviewed the changes, because of time constraints). Do you explain how you found these? Neither value for Colpach seems to match the HAND histogram in Fig. 5, and you do not indicate the values for Wollefsbach .

EZ: Obviously we did not explain this well enough (and note that the histogram of the observations points are given in Figure (3), Figure 5 provides those for the entire catchment). In this exercise we contaminated the HAND value with an error of 2m and plotted the corresponding energy state curve. This curve does considerably mismatch the observations. This corroborates on a) that HAND is a good estimator of depth to groundwater at this point and B) that an error in the estimated depth to groundwater leads to a mismatch between the theoretical state curve and the observed values. So the method is indeed sensitive to depth to groundwater, as correctly stated by GR above. But this opens in fact options for learning, as elaborated in the discussion section.

To better explain this we added the following to the new manuscript. In a further step we contaminated the HAND values of both sites with an error of 2m and plotted the corresponding energy state curves ($z_{HAND} = 18$ m). This curve does considerably mismatch the observations (Figure 6b, c). This corroborates a) that HAND is a good estimator of depth to groundwater at this point and b) that an error in the estimated depth to groundwater leads to a mismatch between the theoretical energy state curve and the observed values. This implies that the observed energy states will also change with changing groundwater surface, as further detailed in the discussion.

GR: l. 686-688: I did not see much evidence for a linear dependence of the free energy on the

degree of saturation (nor did I expect it). Please elaborate. I would like to have some clarification on the determination of the depth to groundwater that separates the wet and the dry branches of your curves.

EZ. Sorry to insist. The energy state curves of the Colpach show for saturation larger than 0.3 a pretty good a constant slope at a given HAND value. This is what I call a linear function and the observed states drop nicely into these linear ranges. An den plotted observations corroborate that efree grows linearly with (efree $= \theta \psi + \theta z$).

The retention function in Figure 1 shows that the matric potential in the Colpach is at the minimum observed saturation of S=0.3 (Figure 6b) equals -2 m. This implies according to Eq. 8 that $e_{free} = - 0.3* \theta s$ 2m $+ 0.3* \theta s$ 20 m $= 2.91$ m and that potential energy is 10 times larger than capillary binding energy. For larger saturations the first term remains rather constant while the second grows linearly with saturation. This is now stated in the manuscript. The growth rate does of course change with HAND.

Thank you very much for the efforts,

Erwin Zehe

References

Zehe, E., Ehret, U., Blume, T., Kleidon, A., Scherer, U., and Westhoff, M.: A thermodynamic approach to link self-organization, preferential flow and rainfall-runoff behaviour, Hydrology And Earth System Sciences, 17, 4297-4322, 10.5194/hess-17-4297-2013, 2013.